# Impacts of ice-nucleating particles on cirrus clouds and radiation derived from global model simulations with MADE3 in EMAC

Christof G. Beer[1], Johannes Hendricks[1], and Mattia Righi[1]

[1]Deutsches Zentrum für Luft- und Raumfahrt (DLR), Institut für Physik der Atmosphäre, Oberpfaffenhofen, Germany

**Correspondence:** Christof Beer (christof.beer@dlr.de)

**Abstract.** Atmospheric aerosols can act as ice-nucleating particles (INPs) and influence the formation and the microphysical properties of cirrus clouds, resulting in distinct climate effects. We employ a global aerosol–climate model, including a two-moment cloud microphysical scheme and a parametrization for aerosol-induced ice formation in cirrus clouds, to quantify the climate impact of INPs on cirrus clouds (simulated period 2001–2010). The model considers mineral dust, soot, crystalline ammonium sulfate, and glassy organics as INPs in the cirrus regime. Several sensitivity experiments are performed to analyse various aspects of the simulated INP-cirrus effect regarding (i) the ice-nucleating potential of the INPs, (ii) the inclusion of ammonium sulfate and organic particles as INPs in the model, and (iii) the model representations of vertical updrafts. The resulting global radiative forcing of the total INP-cirrus effect, considering all different INP-types, assuming a smaller and a larger ice nucleating potential of INPs, to explore the range of possible forcings due to uncertainties in the freezing properties of INPs, is simulated as $-28$ and $-55\ \mathrm{mW\,m^{-2}}$, respectively. While the simulated impact of glassy organic INPs is mostly small and not statistically significant, ammonium sulfate INPs contribute a considerable radiative forcing, which is nearly as large as the combined effect of mineral dust and soot INPs. Additionally, the anthropogenic INP-cirrus effect is analysed considering the difference between present-day (2014) and pre-industrial conditions (1750) and amounts to $-29\ \mathrm{mW\,m^{-2}}$, assuming a larger ice-nucleating potential of INPs. In an additional sensitivity experiment we analyse the effect of highly efficient INPs proposed for cirrus cloud seeding as a means to reduce global warming by climate engineering. However, the results indicate that this approach risks an overseeding of cirrus clouds and often results in positive radiative forcings of up to $86\ \mathrm{mW\,m^{-2}}$ depending on number concentration of seeded INPs. Idealized experiments with prescribed vertical velocities highlight the crucial role of the model dynamics for the simulated INP-cirrus effects, e.g. resulting forcings increase about one order of magnitude ($-42$ to $-340\ \mathrm{mW\,m^{-2}}$) when increasing the prescribed vertical velocity (from 1 to $50\ \mathrm{cm\,s^{-1}}$). The large discrepancy in the magnitude of the simulated INP-cirrus effect between different model studies emphasizes the need for future detailed analyses and efforts to reduce this uncertainty and constrain the resulting climate impact of INPs.

## 1 Introduction

Atmospheric aerosol particles can exert various influences on the climate system. Aerosols change the Earth's radiation budget by directly interacting with solar and terrestrial radiation through absorption and scattering (Boucher et al., 2013; Szopa et al., 2021). Importantly, aerosol particles can also act as cloud condensation nuclei and ice nucleating particles, consequently

influencing the formation of cloud droplets and ice crystals and leading to additional indirect climate effects (Boucher et al., 2013). However, these aerosol-cloud interactions still pose several remaining open questions and are the subject of ongoing research activities (Mülmenstädt and Feingold, 2018; Bellouin et al., 2020; Murray et al., 2021; Szopa et al., 2021).

Especially the knowledge about the effect of ice nucleating particles (INPs) on cirrus clouds and the resulting climate effects are highly uncertain (Kärcher, 2017; Kanji et al., 2017). Cirrus clouds typically exert a warming effect on the global climate due to their strong absorption of outgoing terrestrial radiation that exceeds the reflection of solar radiation (Chen et al., 2000; Gasparini and Lohmann, 2016). INPs contribute to the climate impacts of cirrus by changing the microphysical properties of cirrus clouds. Compared to the homogeneous freezing of liquid aerosols, INPs can initiate heterogeneous nucleation of ice crystals at lower supersaturations with respect to ice (Koop et al., 2000; Hoose and Möhler, 2012). The resulting competition mechanism between heterogeneous and homogeneous nucleation for the available supersaturated water vapour critically depends on the abundance of INPs and their freezing properties and can induce substantial changes in the microphysical properties of clouds, e.g. via changes in ice crystal number concentration and ice crystal sizes, which in turn influence the cloud optical thickness (Kärcher et al., 2006; Gasparini and Lohmann, 2016; Righi et al., 2021). To characterize the global radiative impact of INPs the term "radiative forcing" (RF) is used, which is defined as the net change of the Earth's energy balance (Ramaswamy et al., 2019), i.e. downward shortwave plus upward longwave radiative flux, due to some imposed perturbation (the impact of INPs on cirrus clouds, in this study).

In the past, several global modelling studies have been performed to evaluate the radiative impact induced by INP-effects on cirrus and mixed phase clouds. While most studies agree on the sign of the global INP-effect, i.e. a net negative radiative forcing leading to a cooling of the climate system, there are conflicting results concerning its magnitude. Estimates range from statistically insignificant effects (Hendricks et al., 2011; Gettelman et al., 2012), to negative forcings ranging from a few $-10$ to several $-100\,\mathrm{mW\,m^{-2}}$ (Liu et al., 2012; Zhou and Penner, 2014; Penner et al., 2018; Zhu and Penner, 2020; McGraw et al., 2020; Righi et al., 2021), or even positive forcings of the order of $100\,\mathrm{mW\,m^{-2}}$ (Liu et al., 2009). However, the comparison of different studies is complicated as: (i) different model systems and techniques are used, (ii) different INP species with varying freezing properties are considered, and (iii) different sensitivities and effects are analysed. There are many reasons for this large range of possible climate effects, e.g. the complexity of the involved processes and the necessity for assumptions and parametrizations to represent sub-grid processes at the large-scale global model resolution. Part of the uncertainties in the INP-cirrus effect is related to the assumptions for the still poorly understood ice nucleating properties of INPs. These are the critical saturation ratio ($S_c$) at which the INPs can initiate freezing and the active fraction ($f_{\mathrm{act}}$) of the aerosol particle population that can act as INPs. Typically, mineral dust and soot particles have been considered in global models for ice nucleation in the cirrus regime (Kuebbeler et al., 2014; McGraw et al., 2020; Righi et al., 2021). However, our knowledge on the global INP population is still uncertain, as several recent studies reported a strong ice-nucleating potential of INP species that previously had not been considered for cirrus ice nucleation, e.g. glassy organic particles (Murray et al., 2010; Ladino et al., 2014; Ignatius et al., 2016; Wagner et al., 2017), and crystalline ammonium sulfate (Abbatt et al., 2006; Baustian et al., 2010; Ladino et al., 2014; Bertozzi et al., 2021). Furthermore, it has been shown that the dynamic forcing, induced by the vertical velocities of the air parcels during the freezing process, is crucial for the competition between heterogeneous and homogeneous nucleation

(Kärcher and Lohmann, 2002, 2003; Kärcher et al., 2006). Despite the importance of subgrid-scale variabilities of updraft speeds, global models are not able to resolve these due to their coarse resolution and need to adopt simplified representations, which can introduce additional uncertainties (Kuebbeler et al., 2014; Righi et al., 2020).

We analyse different aspects of INP-induced cirrus modifications in a global aerosol-climate model with the goal to explore the large uncertainties in the understanding of the aerosol-cirrus effect. We employ the atmospheric chemistry general circulation model EMAC (ECHAM/MESSy Atmospheric Chemistry model; Jöckel et al., 2010) including the MESSy (Modular Earth Submodel System) aerosol microphysics submodel MADE3 (Modal Aerosol Dynamics model for Europe, adapted for global applications, third generation; Kaiser et al., 2014; 2019; Beer et al., 2020). The MADE3 aerosol is coupled to (cirrus) clouds via a two-moment cloud-scheme (Kuebbeler et al., 2014) as described by Righi et al. (2020). In addition to mineral dust and soot, glassy organics and crystalline ammonium sulfate particles are represented as INPs in the model as described in detail by Beer et al. (2022).

We investigate the impact of different subsets of the INP population, e.g. the effect of anthropogenic INPs. Also, the impact of ammonium sulfate INPs is evaluated, as Beer et al. (2022) recently reported a potentially large impact of this INP type. We further analyse the influences of different assumptions for the ice nucleating properties of the INPs. In order to explore the sensitivity of INP-cirrus effects to the model dynamics, we perform mechanistic studies by varying the representation of vertical velocities in the model. We discuss possible sources of uncertainties in the simulated INP-effects, e.g. due to the use of model nudging, dependencies on the applied model resolution, and model assumptions regarding the parametrization of INP-cirrus interactions. Additionally, we analyse the effect of highly efficient INPs, which have been proposed to engineer climate and reduce global warming by cirrus cloud seeding (Mitchell and Finnegan, 2009; Gasparini and Lohmann, 2016). By designing the simulation experiments for this analysis according to the study by Gasparini and Lohmann (2016), we improve the comparability and aim to explore the robustness of the resulting quantifications of the INP-cirrus effects presented here by comparing with a similar model study.

This paper is organized as follows. In Sect. 2 we provide an overview on the modelling framework EMAC, including the aerosol submodel MADE3 and its coupling to (cirrus) clouds. Results on the effect of INPs on cirrus clouds and radiation are presented in Sect. 3, regarding the impact of the assumed ice-nucleating properties of INPs, the effect of different INP species, the role of the representation of vertical velocities in the model, and the impact of highly efficient INPs proposed for cirrus cloud seeding. The work presented in this paper is in parts based on the PhD thesis by C. G. Beer (Beer, 2021) and some of the text appeared similarly therein.

## 2 Model description

The EMAC model is a global numerical chemistry and climate simulation system and includes various submodels that describe tropospheric and middle-atmosphere processes. It uses the second version of MESSy to connect multi-institutional computer codes. The core atmospheric model is the ECHAM5 (fifth-generation European Centre Hamburg) general circulation model (Roeckner et al., 2006). In this work we apply EMAC (ECHAM5 version 5.3.02, MESSy version 2.54) in the T42L41 configu-

ration with spherical truncation of T42 (corresponding to a horizontal resolution of about $2.8° × 2.8°$ in latitude and longitude) and 41 non-equidistant vertical layers from the surface to $10\,\text{hPa}$. The simulated time period covers the years 2000 to 2010, while the year 2000 is used as a spin-up and excluded from the evaluation. Most simulations presented here are performed in nudged mode, i.e. model meteorology (temperature, winds and logarithm of the surface pressure) is relaxed towards ECMWF reanalyses (ERA-Interim; Dee et al., 2011) for the simulated time period. The original reanalysis data with a spectral horizontal resolution of T255 ($0.54° × 0.54°$) and a vertical resolution of 60 levels from the ground up to $0.1\,\text{hPa}$ have been re-gridded to the model resolution used in this study. The nudging data have a temporal resolution of 6 hours. Additional sensitivity experiments are performed in free-running mode to analyse the effect of nudging on the results also covering the time period 2001–2010, using prescribed long-term means (2001–2010) of sea-surface temperature and sea-ice concentration from the Met Office Hadley Centre dataset (HadISST; Rayner et al., 2003).

The aerosol microphysics submodel MADE3 simulates different aerosol species in nine log-normal modes that represent different particle sizes and mixing states. The aerosol components in MADE3 are distributed into nine lognormal modes that represent different particle sizes and mixing states. Each of the MADE3 Aitken-, accumulation-, and coarse-mode size ranges includes three modes for different particle mixing states: particles fully composed of water-soluble components, particles mainly composed of insoluble material (i.e. insoluble particles with only very thin coatings of soluble material), and mixed particles (i.e. soluble material with inclusions of insoluble particles). A detailed description of MADE3 and its application and evaluation as part of EMAC is presented in Kaiser et al. (2014, 2019). The optical properties of aerosols and clouds are calculated in the EMAC submodels AEROPT and CLOUDOPT (Dietmüller et al., 2016), respectively, according to input from OPAC (optical properties for aerosols and clouds; Hess et al., 1998). The OPAC package uses basic optical properties from Koepke et al. (1997).

The EMAC-MADE3 setup applied here is in large parts based on the setup described in Righi et al. (2021). We use the recent CMIP6 (Coupled Model Intercomparison Project, phase 6) emission inventory for anthropogenic and biomass burning emissions of aerosols and aerosol precursor species (van Marle et al., 2017; Hoesly et al., 2018) for the year 2014. Prescribed emission data are provided in a horizontal resolution of $0.5° × 0.5°$. The re-gridding to the actual model grid is performed during the model simulation using the algorithm NCREGRID (Jöckel, 2006). Since our study focuses on the radiative effects of aerosols on (cirrus) clouds, the concentrations of radiatively active gases other than water vapor (i.e., $CO_2$, $CH_4$, $N_2O$, $O_3$, and chlorofluorocarbons) are prescribed by means of global distributions. In order to simulate the effect of anthropogenic emissions on the INP-cirrus effects, a pre-industrial simulation is performed considering year 1750 emissions for anthropogenic and biomass burning sources, instead of the 2014 (present-day) ones. Mineral dust emissions are calculated according to the online emission scheme of Tegen et al. (2002), as described and evaluated in Beer et al. (2020). In order to isolate the effect of anthropogenic emissions, mineral dust emissions and all other natural emissions in the model (i.e. biogenic emissions, volcanic emissions, $NO_x$ emissions from lightning, emissions of SOA precursors, dimethyl sulfide (DMS) emissions, and wind-driven sea-spray emissions) are kept constant at their present-day value in all simulations performed here.

In this study, EMAC-MADE3 is employed in a coupled configuration which includes a two-moment cloud microphysical scheme based on Kuebbeler et al. (2014), employing a parametrization for aerosol-driven ice formation in cirrus clouds

following Kärcher et al. (2006). The Kärcher et al. (2006) scheme considers the competition between various ice formation
mechanisms for the available supersaturated water vapour, i.e. homogeneous freezing of solution droplets, deposition and im-
mersion nucleation induced by INPs, and the growth of preexisting ice crystals. By way of simplification the scheme does
not represent the whole freezing spectrum (from the freezing onset to the homogeneous freezing threshold) but considers
one singular point in the spectrum, e.g. corresponding to the freezing onset. In each of the heterogeneous freezing modes the
ice-nucleating properties of the INPs are represented by two parameters, namely the active fraction ($f_{\mathrm{act}}$) of ice nucleating
particles, which actually lead to the formation of ice crystals, and the critical supersaturation ratio with respect to ice ($S_c$),
at which the freezing process is initiated. In addition to choosing $f_{\mathrm{act}}$ representative of the freezing onset, we also consider
a larger $f_{\mathrm{act}}$ value representing the mid point between the freezing onset and the homogeneous freezing threshold. With this
sensitivity experiment we aim to explore the range of possible forcings due to uncertainties in the freezing properties of INPs
also accounting for increasing activated fractions during the freezing process, which might be more representative for the total
number of activated particles. We discuss possible sources of uncertainties influencing the simulated results due to the applied
cirrus cloud parametrization in Sect. 4.4. The model setup has been extensively tuned and evaluated with respect to various
cloud and radiation variables by Righi et al. (2020) with further model improvements described in Righi et al. (2021).

The aerosol-driven ice formation scheme includes additional model developments to represent crytalline ammonium sulfate
(AmSu) and glassy organic particles (glPOM), in addition to mineral dust (DU) and soot (BC), as ice nucleating particles.
The model is able to distinguish between aviation soot and soot from other sources (Righi et al., 2021), however, we choose
the same freezing properties for these two soot classes in this study due to the uncertain freezing properties of aviation BC.
The model improvements regarding glassy organics and crystalline ammonium sulfate were described in detail by Beer et al.
(2022) and are briefly summarized here. Importantly, the particle phase state is tracked for glassy organics and crystalline
ammonium sulfate, as only the respective glassy or crystalline phase facilitates ice nucleation. Natural secondary organic
aerosol (SOA) precursor emissions (e.g. isoprene, monoterpenes, and other volatile organic compounds) according to Guenther
et al. (1995) are considered for the glassy organics tracer in the model. We use a simplified representation for the glass-transition
temperature $T_g$, depending on the relative humidity (RH). For temperatures $T < T_g$ liquid SOA particles are transformed into a
semi-solid, glassy state. $T_g(\mathrm{RH})$ for the SOA proxy citric acid is calculated according to Baustian et al. (2013). For crystalline
ammonium sulfate, a dedicated phase transition formulation via a hysteresis process depending on the relative humidity history
is considered in the model. Efflorescence of aqueous ammonium sulfate particles occurs at a lower relative humidity than the
deliquescence of ammonium sulfate crystals. The number concentrations of potential INPs are calculated for the different ice
formation modes in the mixed-phase and the cirrus regime according to the procedure described in Righi et al. (2020), and
adapted by Beer et al. (2022) considering the additional INP types ammonium sulfate and glassy organics. Different mixing
states of particles are taken into account for the simulated ice-nucleation processes, e.g. the model distinguishes between
immersion freezing of mixed mineral dust particles and deposition freezing of insoluble mineral dust. The freezing properties,
i.e. values for $f_{\mathrm{act}}$ and $S_c$, for the different INP species are assumed according to Beer et al. (2022), and are summarized in
Table 1.

**Table 1.** Freezing properties of ice nucleating particles in the cirrus regime assumed in this study, i.e. critical supersaturation $S_c$ and activated fraction $f_{\text{act}}$ at the freezing onset. $S_i$ is the supersaturation with respect to ice. As an alternative to $f_{\text{act}}$ values representative of the freezing onset, $f_{\text{act}}$ values in the centre of the activation spectrum are assumed for a sensitivity experiment, representing the midpoint between the onset and the homogeneous freezing threshold.

| Freezing mode | | $S_c$ | $f_{\text{act}}$ at onset $S_c$ | $f_{\text{act}}$ at central $S_i$ | Reference |
|---|---|---|---|---|---|
| DU deposition | $T \leq 220$ K | 1.10 | $\exp[2\,(S_i - S_c)] - 1$ | | Möhler et al. (2006) |
| | $T > 220$ K | 1.20 | $\exp[0.5\,(S_i - S_c)] - 1$ | | |
| AmSu | | 1.25 | 0.001 | 0.05 | Ladino et al. (2014) |
| glPOM | | 1.30 | 0.001 | 0.06 | Ignatius et al. (2016) |
| DU immersion | | 1.35 | 0.01 | 0.065 | Kulkarni et al. (2014) |
| BC | | 1.40 | 0.001 | 0.006 | Kulkarni et al. (2016) |

In a set of sensitivity experiments we analyse the influence of variations in the updraft velocities on the INP-cirrus effects. In the reference case vertical velocities are parametrized in the model to account for the sub-grid scale variability of updraft velocities, that cannot be resolved with the coarse model resolution. As described in Righi et al. (2020), the total updraft speed is represented as the sum of the large-scale vertical velocity, a turbulent component which is proportional to the square root of the turbulent kinetic energy (TKE; Kärcher and Lohmann, 2002), and an additional term accounting for orographic gravity waves calculated by the orographic gravity wave submodel (OROGW) in EMAC, based on the parametrization by Joos et al. (2008). The resulting global distribution of simulated vertical velocities (including also the single components) can be found in Righi et al. (2021). To further investigate these dynamic influences on the INP-cirrus effects, we also consider an idealized representation of the vertical velocity. We prescribe constant values of the vertical velocity across the whole globe in the range from 1 to 50 $\text{cm}\,\text{s}^{-1}$ to explore the full range updraft speeds typically occurring in the atmosphere (Podglajen et al., 2016; Barahona et al., 2017). This also enables the investigation of INP effects in regimes not covered by the updraft parametrization considering TKE and orographic gravity waves mentioned above. An overview of the different simulation experiments presented here is provided in Table 2.

## 3  Model resolution dependencies

As described in the previous section, the simulations presented in this study have been performed using the T42L41 model resolution, representing a horizontal resolution of about $2.8° \times 2.8°$ in latitude and longitude and 41 vertical layers. This model resolution is different from the one applied in previous studies by Beer et al. (2020) and Beer et al. (2022), i.e. T63L31 (corresponding to a horizontal resolution of $2.8° \times 2.8°$ and 31 vertical layers). For the present study focusing on the climate impact of INPs, we rely on the extensive model tuning of cloud and radiation properties by Righi et al. (2020, 2021), applying the T42L41 resolution. Additionally, performing the large number of simulations presented here, would not be feasible applying

**Table 2.** Summary of the global model simulations performed in this study. Each experiment considers the difference between two simulations, i.e. one with heterogeneous freezing on INPs and one without, i.e. only homogeneous freezing. The homogeneous freezing reference cases are marked in bold. An exception are the simulations that consider AmSu (glPOM) in addition to DU and BC (simulations F-CEN-DBA, F-CEN-DBG; see below), which consider the simulation with DU and BC as the reference case (simulation F-CEN-DB). The freezing properties ($S_c$, $f_{act}$) of INPs are also summarized in Table 1. Every simulation includes an extra spin-up year which is not considered for the analysis. The nudged simulations use meteorological reanalysis data for the period 2001–2010. The pre-industrial simulation considers year 1750 emissions for anthropogenic and biomass burning sources, instead of the 2014 (present-day) ones. All simulations cover a 10-year period (2001-2010).

| Name | INP species | $S_c$ | $f_{act}$ | Vertical velocity [$\mathrm{cm\,s^{-1}}$] | Dynamics | Emissions |
|---|---|---|---|---|---|---|
| F-ONS | DU, BC, AmSu, glPOM | Table 1 | onset | online | nudged | present-day |
| F-CEN | DU, BC, AmSu, glPOM | Table 1 | central | online | nudged | present-day |
| F-CEN-DB | DU, BC | Table 1 | central | online | nudged | present-day |
| F-CEN-DBA | DU, BC, AmSu | Table 1 | central | online | nudged | present-day |
| F-CEN-DBG | DU, BC, glPOM | Table 1 | central | online | nudged | present-day |
| **HOM** | **none (only hom. freezing)** | **—** | **—** | **online** | **nudged** | **present-day** |
| F-CEN-PI | DU, BC, AmSu, glPOM | Table 1 | central | online | nudged | pre-industrial |
| **HOM-PI** | **none (only hom. freezing)** | **—** | **—** | **online** | **nudged** | **pre-industrial** |
| F-CEN-V1 | DU, BC, AmSu, glPOM | Table 1 | central | 1 | nudged | present-day |
| F-CEN-V5 | DU, BC, AmSu, glPOM | Table 1 | central | 5 | nudged | present-day |
| F-CEN-V10 | DU, BC, AmSu, glPOM | Table 1 | central | 10 | nudged | present-day |
| F-CEN-V20 | DU, BC, AmSu, glPOM | Table 1 | central | 20 | nudged | present-day |
| F-CEN-V50 | DU, BC, AmSu, glPOM | Table 1 | central | 50 | nudged | present-day |
| **HOM-V$x$[a]** | **none (only hom. freezing)** | **—** | **—** | $x \in \{1, 5, 10, 20, 50\}$ | **nudged** | **present-day** |
| SEED-0.5 | Seed INPs, $0.5\,\mathrm{L^{-1}}$ | 1.05 | 1.0 | online | nudged | present-day |
| SEED-1 | Seed INPs, $1.0\,\mathrm{L^{-1}}$ | 1.05 | 1.0 | online | nudged | present-day |
| SEED-10 | Seed INPs, $10\,\mathrm{L^{-1}}$ | 1.05 | 1.0 | online | nudged | present-day |
| SEED-100 | Seed INPs, $100\,\mathrm{L^{-1}}$ | 1.05 | 1.0 | online | nudged | present-day |
| **HOM** | **none (only hom. freezing)** | **—** | **—** | **online** | **nudged** | **present-day** |
| F-CEN-V20-F | DU, BC, AmSu, glPOM | Table 1 | central | 20 | free | present-day |
| **HOM-F** | **none (only hom. freezing)** | **—** | **—** | **20** | **free** | **present-day** |

[a] Baseline simulations with only homogeneous freezing have been performed for each vertical velocity.

the T63L31 model setup, that would require considerably more computational resources due to the increased horizontal model resolution. However, the lower horizontal resolution used in this study also has some drawbacks, that are discussed in the

following. As described in Beer et al. (2020), a lower horizontal model resolution can introduce a positive bias for aerosol number concentrations compared to observations, that is most pronounced in the upper troposphere. In Fig. S1 in the Supplement, we present a comparison of aerosol number concentrations for the different model resolutions, which complements Fig. 5 of Beer et al. (2020) with the results for the T42L41 resolution applied in the present study. Aerosol number concentrations above $400\,\text{hPa}$ are about a factor of 2 to 3 larger in the T42L41 resolution compared to T63L31. As discussed in Beer et al. (2020), this could be related to overestimated upward transport, possibly in convective plumes, or underestimation of aerosol scavenging through a too low efficiency of the wet deposition processes in the model.

Figure 1 shows the global distributions of the simulated INP number concentrations as shown in Beer et al. (2022, Fig. 5) but for the T42L41 model resolution. While the main findings and conclusions discussed in Beer et al. (2022) still hold for the decreased (increased) horizontal (vertical) model resolution, the simulated INP number concentrations are about a factor of 2 larger compared to the INP concentrations presented in Beer et al. (2022) for the T63L31 model resolution. The effect of a lower horizontal model resolution has to be considered for the interpretation of the resulting climate forcings due to INPs as presented here, and would likely result in larger INP-effects due to the increased concentration of INPs. Notably, the model resolution can also influence cloud formation in the model, e.g. via changes in the simulated vertical velocity, which acts as a driver for the supersaturation and hence the ice-nucleation processes in the model can be influenced by the applied model resolution. In general, the differences in cloud frequency and vertical velocities between the T42L41 and T63L31 model resolutions are smaller compared to the differences in INP numbers, i.e. mostly below $50\,\%$ in the cirrus regime (data not shown). Nonetheless, the applied model resolution can influence the simulated INP-cirrus effects and this impact should be the focus of future studies.

## 4 Results and discussion

### 4.1 Simulated INP-cirrus effects

The results presented here describe the effect of INPs on cirrus clouds and radiation, calculated as the difference between simulations with and without heterogeneous freezing induced by INPs. These differences are often small and sparsely distributed, which makes them difficult to quantify, due to the low magnitude of the effect with respect to the internal model variability. Therefore, the analysis is performed considering differences of aggregated quantities at the global and regional level and the typical cirrus altitudes (above $400\,\text{hPa}$) to facilitate the interpretation of the results. However, one has to keep in mind, that these aggregated results can represent a superposition of different local regimes and effects. The radiative forcings reported here explicitly consider the impact of cloud adjustments, as we are employing an aerosol-cloud coupled model. The RF values presented in the following can therefore be regarded as approximations of effective radiative forcings (ERF), although the use of model nudging may tend to suppress some feedbacks (see Sect. 4.3). Therefore, we use the term ERF in the following.

In Fig. 2, the effect of all INPs (i.e. DU, BC, BCair, AmSu, and glPOM) on different cloud and radiation variables is shown. Results for top-of-the-atmosphere radiative forcings (ERF; all-sky, cloudy-sky, longwave, and shortwave) are represented as absolute differences in units of $\text{mW}\,\text{m}^{-2}$ (Fig. 2a-e), while relative differences (in %) are shown for the other variables (Fig. 2f-

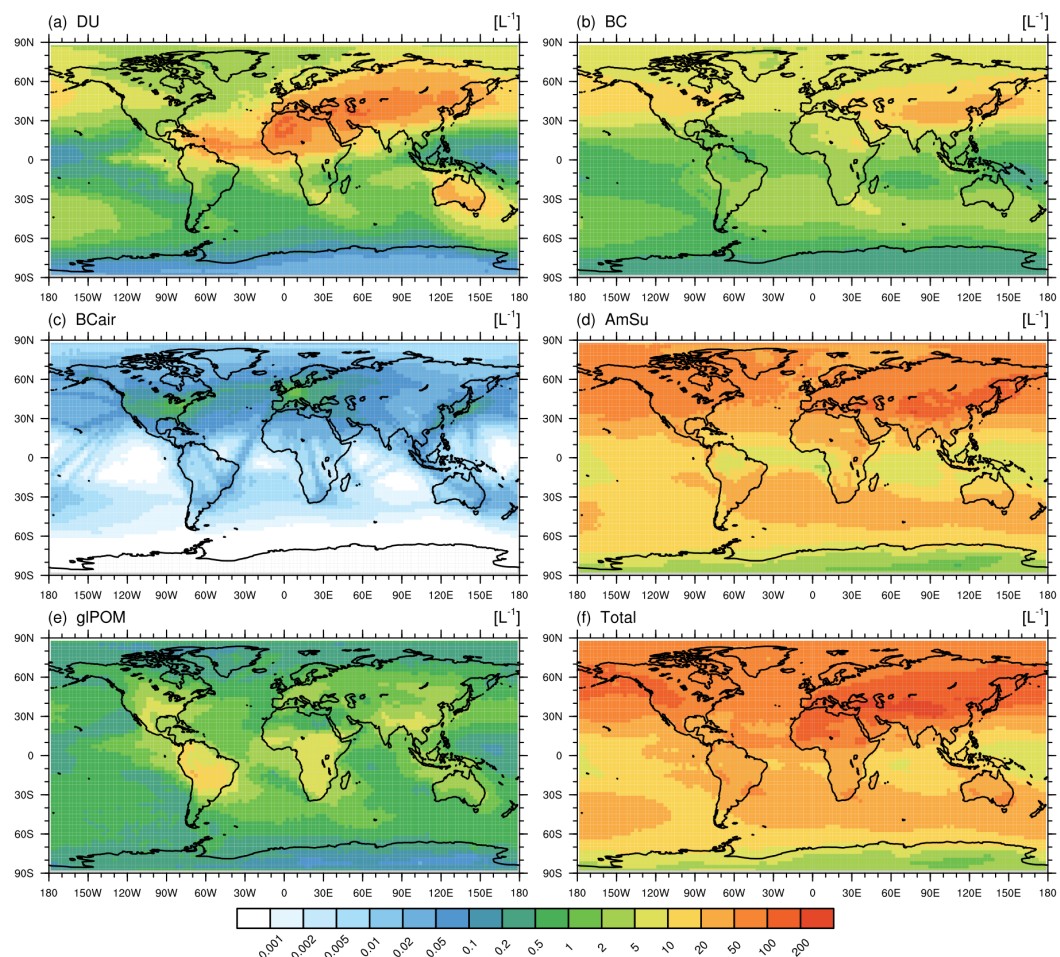

**Figure 1.** Global distribution of the simulated number concentrations of different INPs per litre ($L^{-1}$) inside cirrus clouds (selecting only grid boxes with cirrus occurrence) considering the multi-year average over the simulation period (2001–2010) and over all vertical levels; similar to Fig. 5 of Beer et al. (2022) but for the T42L41 model resolution applied here. Shown are mineral dust (DU; a), black carbon (BC; b), black carbon from aviation (BCair; c), ammonium sulfate (AmSu; d), glassy organics (glPOM; d), and total INP concentrations (f). Cirrus conditions are selected according to thresholds for simulated ambient temperature ($T < 238\,\mathrm{K}$) and ice water content (IWC $> 0.5\,\mathrm{mg\,kg^{-1}}$) in every grid box using the 11 h output frequency. The number concentrations of potential INPs are weighted with ice-active fractions at ice supersaturations of $S_i = 1.4$ from laboratory measurements (see Table 1 in Beer et al.; 2022).

i), i.e. ice crystal number concentration (ICNC), homogeneous freezing fraction, total water (as the sum of water vapour and ice water), and cloud occurrence frequency. For each quantity the global and regional, zonal (Southern Hemisphere extra-tropics, tropics, Northern Hemisphere extra-tropics; see the caption of Fig. 2 for details) differences are shown. The freezing properties of INPs ($S_c$, $f_{\mathrm{act}}$) are representative of the freezing onset (see Table 1). The statistical confidence of the calculated anomalies is assessed employing a paired-sample Student's t-test, with the null hypothesis that the annual mean values of a given quantity

(e.g. the radiative forcing) are identical in the two simulations (with and without heterogeneous freezing). The response of the t-test is represented in terms of confidence levels (in %), i.e. $100\,(1-p)$, where $p$ is the p-value. The results are regarded as statistically significant for confidence levels larger than 95 % ($p < 0.05$).

In general, cirrus clouds have a pronounced longwave warming effect (absorption of terrestrial radiation), which together with the smaller shortwave cooling (reflection of solar radiation) results in a net warming of the atmosphere (e.g. Chen et al., 2000; Gasparini and Lohmann, 2016). This global warming effect is strongly enhanced during night times due to the missing shortwave cooling effect. The presence of INPs typically results in a thinning of cirrus clouds and in turn a reduced cirrus warming, i.e. a cooling effect (e.g. Kuebbeler et al., 2014; Penner et al., 2018; McGraw et al., 2020).

Our model results indeed show a negative radiative forcing due to heterogeneous freezing on INPs, i.e. a global cooling of the climate system with a ERF of $-28 \pm 22\ \mathrm{mW\,m^{-2}}$, considering the confidence interval for the 95 % confidence level (Fig. 2a). This cooling effect is largest in the extratropics, i.e. ERF of $-52 \pm 44\ \mathrm{mW\,m^{-2}}$ and $-41 \pm 20\ \mathrm{mW\,m^{-2}}$ for the Southern and Northern Hemisphere, respectively. The total ERF is the sum of the shortwave and longwave forcings (Fig. 2b,c) and the cooling effect of INPs is mostly related to a negative longwave ERF. This means that the INPs act to decrease the longwave warming of the cirrus clouds, resulting in a longwave cooling effect, while the opposite is true in the shortwave (albeit with lower statistical significance). Ice crystal numbers, homogeneous freezing fraction, and total water mass are reduced with respect to the pure homogeneous freezing case (Fig. 2f–h), whereas the occurrence frequency of clouds is mostly enhanced (Fig. 2i). This analysis is based on global or zonal averages (tropics, extratropics). However, large regional variations can occur due to the different geographical distribution of INPs and cirrus clouds, as shown in Beer et al. (2022). Additionally, large spatial variations in the vertical velocity simulated by the model can lead to regional differences in the INP-cirrus effects. The regional variations in the simulated INP-cirrus effects are shown in Fig. S2 in the Supplement.

In theory, the presence of INPs results in a competition between heterogeneous freezing (at relatively low ice-supersaturation) and homogeneous freezing (at higher supersaturation). This typically leads to a reduction in ice crystal numbers, and the formation of larger ice crystals compared to homogeneous freezing, as shown, e.g., in process-model studies by Kärcher et al. (2006) and Kärcher (2022). This reduction of ice crystal numbers is also visible in the global simulations (Fig. 2f). The decrease of total water (water vapour and ice) mass indicates an increased sedimentation of the larger crystals resulting in a thinning of cirrus clouds, as also shown in other global modelling studies (Kuebbeler et al., 2014; McGraw et al., 2020). On the other hand, the presence of INPs can also increase the occurrence frequency of cirrus clouds, as heterogeneous freezing occurs earlier than homogeneous freezing (at lower critical supersaturations with respect to ice), resulting in an increased cloud occurrence (Fig 2i). Differences between tropics and extra-tropics can be explained by the geographical distribution of the simulated effects (Fig. S2 in the Supplement). In the extra-tropics, regions of positive and negative changes in total water (Fig. S2h) lead to small and not significant effects in these regions. For the cloud frequency (Fig. S2i), positive and negative changes cancel each other in the tropics resulting in the largest effects in the extra-tropics, where the increase in cloud frequency is most pronounced. The vanishing ERF in the tropics may also be due to the strong influence of convection in that region, resulting in enhanced updrafts that are more favourable for homogeneous freezing (Kärcher et al., 2006).

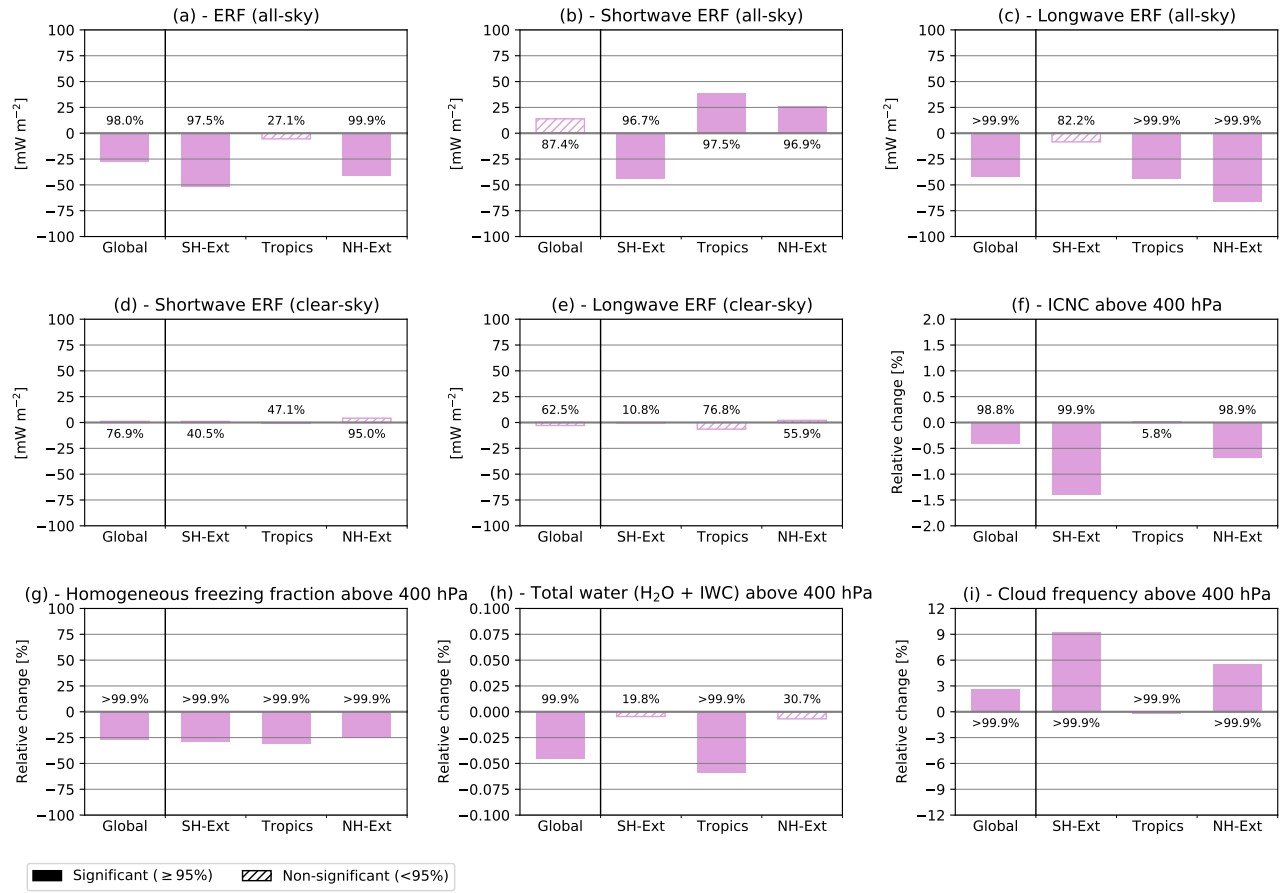

**Figure 2.** Total INP-cirrus effect considering all INPs calculated from the difference between a simulation including these INPs and a simulation with only homogeneous freezing, considering the multi-year average over the simulated period (2001–2010). Freezing properties of INPs are assumed according to freezing onset values for the activated fractions $f_{\mathrm{act}}$, see Table 1). Global and latitude-specific, regional differences are shown for (a) total all-sky, (b) all-sky shortwave, (c) all-sky longwave, (d) clear-sky shortwave, (e) clear-sky longwave top-of-the-atmosphere ERFs in units of $\mathrm{mW\,m^{-2}}$. Panels (f-i) depict relative changes in the all-sky ICNC, fraction of homogeneously formed ice crystals, total water (as the sum of water vapour and ice water), and cloud occurrence frequency, all spatially averaged above the $400\,\mathrm{hPa}$ level and over cloudy and cloud-free grid boxes. Global and latitude-specific, regional values are shown for the Southern Hemisphere extra-tropics ($90°$–$30°$ S), tropics ($30°$ S–$30°$ N), and Northern Hemisphere extra-tropics ($30°$–$90°$ N). Confidence levels (in %) with respect to the model inter-annual variability are calculated according to a two-tailed Student's t-test and are shown for each bar. Significant and non-significant results are represented by filled and hatched bars, respectively.

The results presented here indicate a thinning of cirrus clouds in the presence of INPs due to fewer ice crystals and stronger sedimentation, which in turn leads to a cooling effect. The regions of strong longwave cooling also coincide with regional

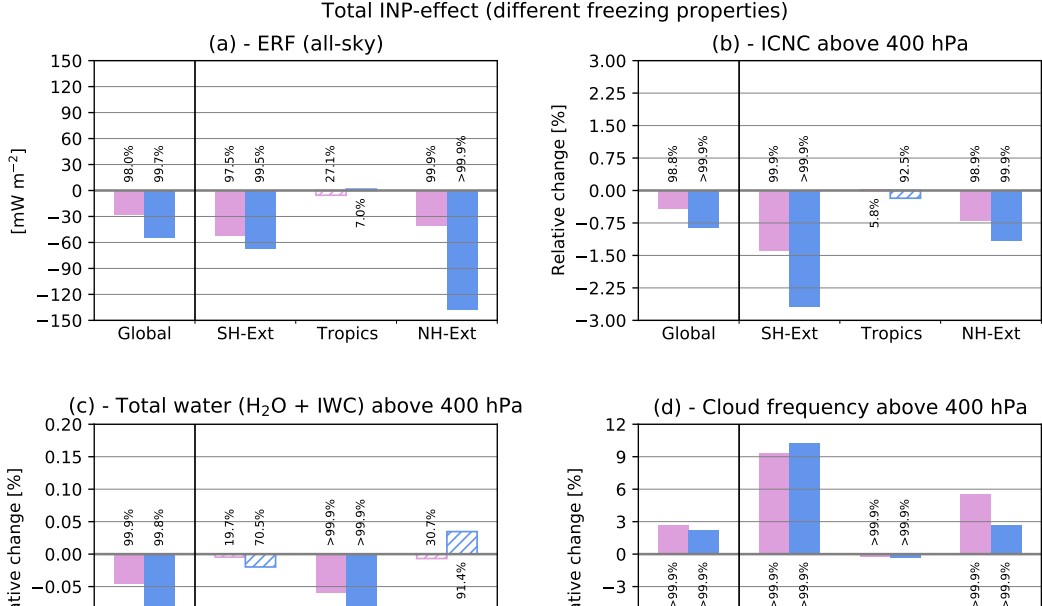

**Figure 3.** As in Fig. 2, but showing the total INP-cirrus effect considering two values for the activated fraction of INPs, i.e. freezing onset values (pink bars, as in Fig. 2), and larger $f_{act}$ values in the centre of the freezing spectrum (blue bars, see Table 1). Global and latitude-specific, regional differences are shown for (a) total all-sky top-of-the-atmosphere ERF in units of $\mathrm{mW\,m^{-2}}$ and relative changes in (b) all-sky ICNC, (c) total water, and (d) cloud occurrence frequency, spatially averaged above the 400 hPa level and considering multi-year averages over the simulated period (2001–2010).

reductions in cloud occurrence (see Fig. S1 in the Supplement). On the other hand, this cooling is weakened by an increase in cloud frequency in other regions, possibly due to more frequent cloud formation or increased cloud lifetimes in the presence of INPs, as a result of the INPs initiating ice formation earlier, i.e. at lower critical supersaturations, compared to homogeneous freezing. Both pathways, i.e. a decrease or an increase in cirrus cloud occurrence due to INPs, have been reported in previous modelling studies depending on the ambient atmospheric conditions and the availability and ice-nucleating properties of INPs (Kuebbeler et al., 2014; Gasparini and Lohmann, 2016; McGraw et al., 2020). Notably, this contributes to the challenge in quantifying the radiative impacts, due to the high variability of the different effects. On the global scale, the combination of the effects mentioned above results in a net negative ERF due to heterogeneous freezing (Fig. 2a). Regional differences are possibly related to variations in INP concentrations as described and presented in Beer et al. (2022). Additionally, regional variations in the vertical updrafts and cooling rates, as shown in Righi et al. (2021), impact the described results.

### 4.1.1 Sensitivity to freezing properties of INPs

In order to explore the uncertainties related to the range of freezing properties (i.e. prescribed values for $S_c$ and $f_{act}$ in the ice nucleation scheme), two cases are analysed (see also Table 1): (i) freezing properties representative of freezing onset conditions (as in the previous section), i.e. low values for $f_{act}$, and (ii) freezing properties in the centre of the freezing spectrum, i.e. larger $f_{act}$ values (midpoint between freezing onset and homogeneous freezing threshold), which might be more representative for the total number of activated particles. An enhancement of the freezing properties, using larger values of $f_{act}$ results in

similar features as for freezing-onset conditions, albeit with larger effects and higher statistical confidence, e.g. global ERF of $-55 \pm 31$ mW m$^{-2}$ (Fig. 3). Typically, the enhanced freezing efficiency of INPs leads to about a factor of 2 larger INP-cirrus effects, e.g. regarding the global ERF (Fig. 3a), or the change in ICNC (Fig. 3b). In addition to the variables shown in Fig. 3, the full set of variables from Fig. 2 is shown in Fig. S3 in the Supplement. Also, the geographical distributions for the increased $f_{act}$-case are shown in Fig. S4 in the Supplement.

In general, the negative sign of the simulated global forcing, i.e. a global cooling, as a result of cloud modifications due to heterogeneous freezing is in line with previous global model studies (e.g. Liu et al., 2012; Kuebbeler et al., 2014; Wang et al., 2014; Penner et al., 2018). However, the simulated global radiative forcings of $-28$ to $-55$ mW m$^{-2}$ (considering the two different assumptions for the INP freezing properties) are smaller compared to several other studies that often simulate radiative effects of the order of $-100$ mW m$^{-2}$ (e.g. Penner et al., 2009; Liu et al., 2012; Wang et al., 2014; McGraw et al.,

2020). On the other hand, some other studies also simulate small, mostly not significant INP-cirrus effects (Hendricks et al., 2011; Gettelman et al., 2012). However, a direct comparison of different model results is difficult due to model differences in the representation of cirrus clouds, INPs and freezing mechanisms, e.g. often different types of INPs (with different freezing properties) were assumed and different effects were analysed.

### 4.1.2 Impacts of different INP-species

In the following, INP-cirrus effects considering different subsets of INP-types are analysed. We choose the case with larger $f_{act}$ values for this analysis (blue bars in Fig. 3), as the larger effects and increased statistical significance facilitates the comparisons. The effect of mineral dust and soot INPs is shown in Fig. 4. The trends are similar to the case where all INP-types can initiate freezing but the magnitude of the effects are often smaller. For example, the radiative effect is reduced by a factor of about two, i.e. a global ERF of $-27 \pm 23$ mW m$^{-2}$, compared to $-55 \pm 31$ mW m$^{-2}$ for the effect of all INPs.

The INP-cirrus effect due to including ammonium sulfate in addition to mineral dust and soot INPs results in an additional global ERF of $-22 \pm 17$ mW m$^{-2}$ (Fig. 5). This is calculated as the difference between a simulation including AmSu, DU and BC INPs, and a simulation including only heterogeneous freezing on DU and BC. The size of the ammonium sulfate effect corresponds almost to the size of the global effect of mineral dust and soot. The large impact of ammonium sulfate is related to the large simulated INP concentrations in the cirrus regime as shown in Beer et al. (2022). Notably, the ammonium sulfate

radiative effect even exceeds that of mineral dust and soot on the Northern Hemisphere, i.e. $-83 \pm 33$ mW m$^{-2}$ (Fig. 5a) compared to $-68 \pm 14$ mW m$^{-2}$ (Fig. 4a), resulting from the large number concentrations of ammonium sulfate INPs in

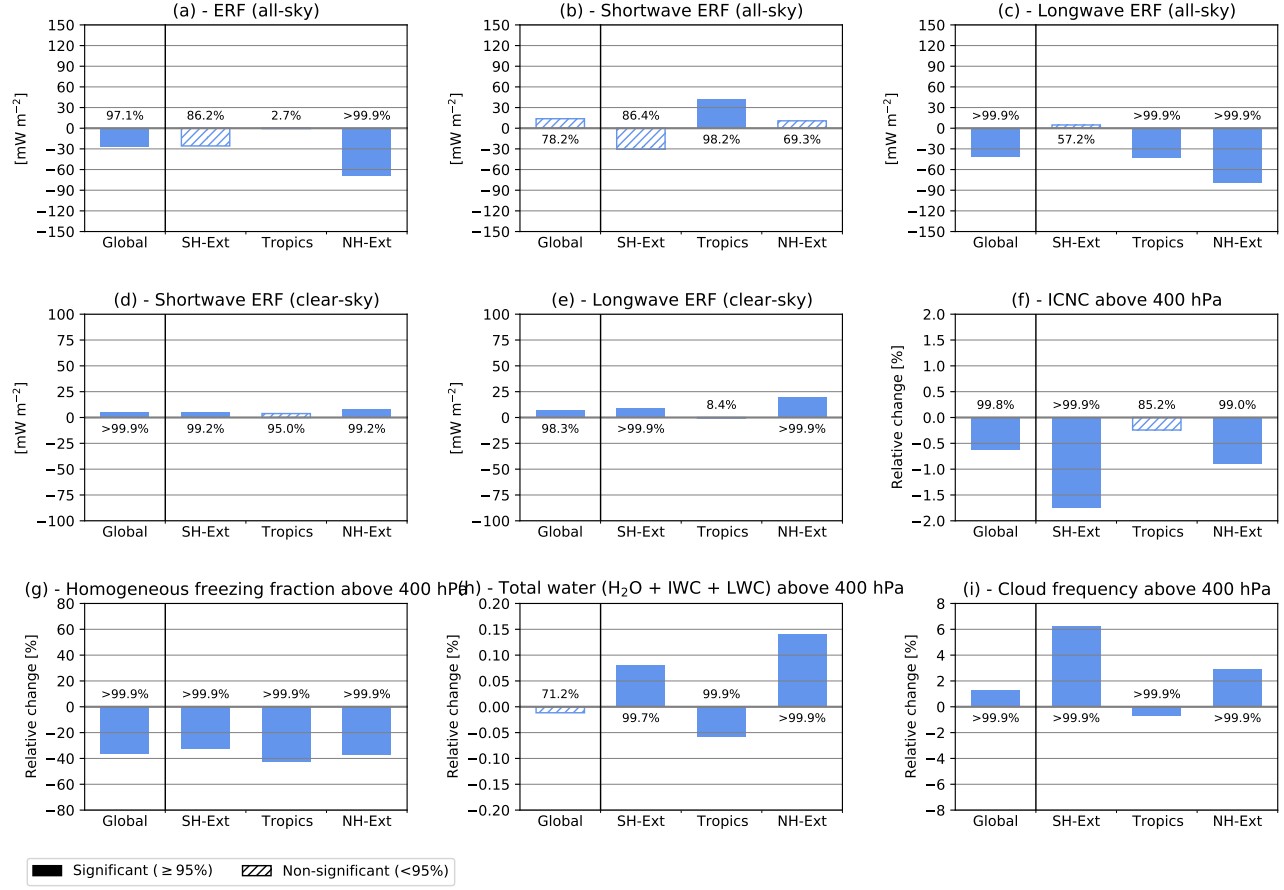

**Figure 4.** As in Fig. 2, but showing the INP-cirrus effect induced by mineral dust and soot INPs only, considering the central $f_{act}$ value, calculated from the difference between a simulation including these INPs and a simulation with only homogeneous freezing. Global and latitude-specific, regional differences are shown for (a) total all-sky, (b) all-sky shortwave, (c) all-sky longwave, (d) clear-sky shortwave, (e) clear-sky longwave top-of-the-atmosphere ERFs. Panels (f-i) depict relative changes in the all-sky ICNC, fraction of homogeneously formed ice crystals, total water (as the sum of water vapour and ice water), and cloud occurrence frequency, all spatially averaged above the 400 hPa level, considering multi-year averages over the simulated period (2001–2010).

that region (Beer et al., 2022). Notably, the ammonium sulfate effect results in a reduced cloud frequency on the Northern Hemisphere (Fig. 5) compared to an increase for the case of mineral dust and soot (Fig. 4). This indicates that the effect of more frequent cirrus formation due to earlier ice nucleation in the presence of INPs is already exhausted by the heterogeneous freezing on mineral dust and soot INPs. The addition of ammonium sulfate INPs further suppresses homogeneous freezing and results in increased sedimentation of ice crystals and lower cirrus occurrence frequencies. In contrast to ammonium sulfate, the effect of glassy organic INPs is not significant for most of the considered variables and latitude regions (see Fig. S5 in the

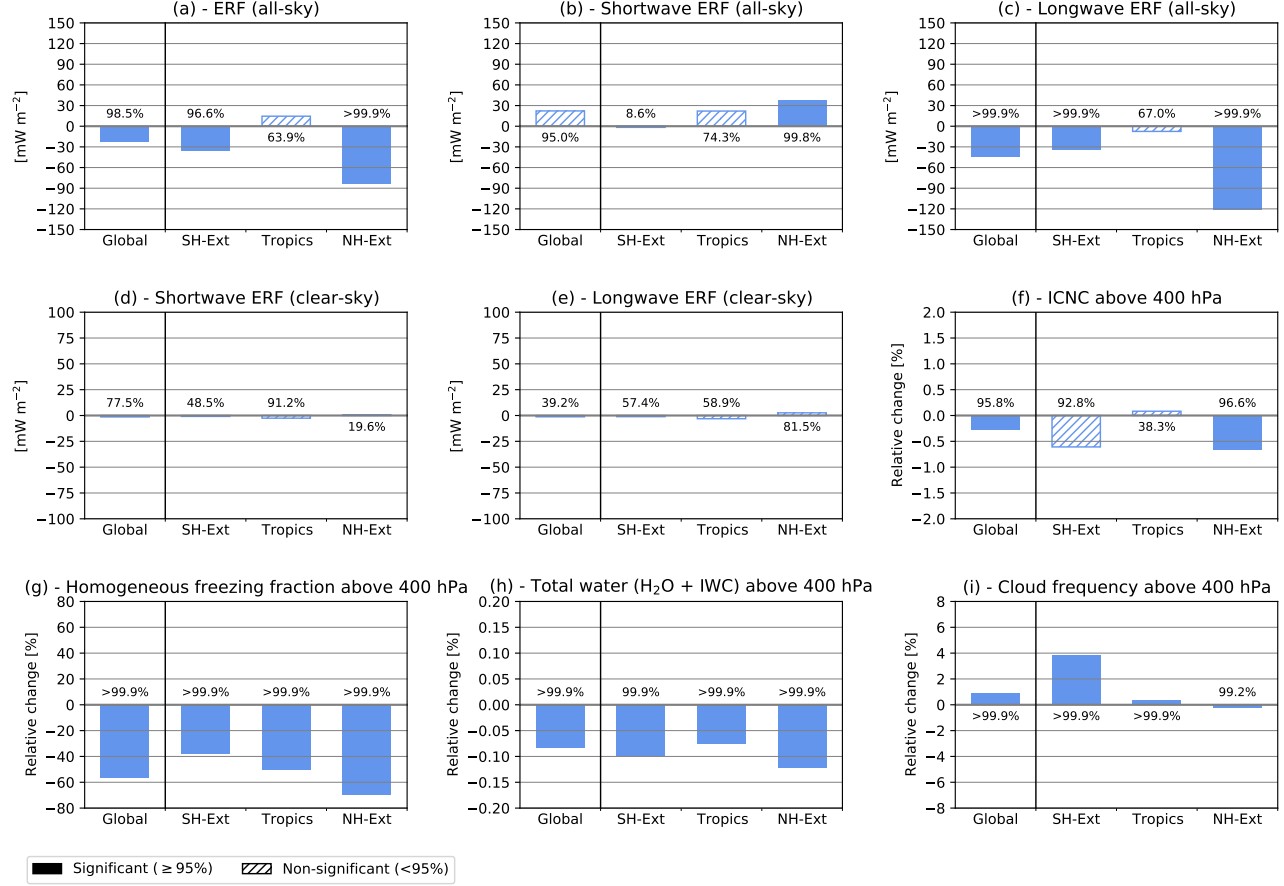

**Figure 5.** As in Fig. 4, but showing the INP-cirrus effect induced by crystalline ammonium sulfate INPs, calculated from the difference between a simulation including AmSu, DU and BC INPs, and a simulation including only heterogeneous freezing on DU and BC. Note the different reference case with respect to Fig. 4. Global and latitude-specific, regional differences are shown for (a) total all-sky, (b) all-sky shortwave, (c) all-sky longwave, (d) clear-sky shortwave, (e) clear-sky longwave top-of-the-atmosphere ERFs. Panels (f-i) depict relative changes in the all-sky ICNC, fraction of homogeneously formed ice crystals, total water (as the sum of water vapour and ice water), and cloud occurrence frequency, all spatially averaged above the 400 hPa level, considering multi-year averages over the simulated period (2001–2010).

Supplement). This is in line with the low concentrations of simulated glassy organic INPs in cirrus clouds described in Beer et al. (2022). A further sensitivity study with an increased number of glassy organic INPs (by choosing a different glassy SOA 310    proxy for calculating the glass transition temperature as described in Beer et al., 2022) also results in no significant impacts of glassy organic INPs (data not shown).

### 4.1.3 Effect of anthropogenic INPs

In this section, the potential influences of anthropogenic INPs on cirrus clouds and climate are analysed. Anthropogenic activities, e.g. the combustion of (sulfur-containing) fossil fuels and the use of ammoniacal fertilizers, mainly affect the atmospheric concentrations of soot, $SO_4$, and $NH_4$, consequently influencing the concentrations of soot and ammonium sulfate INPs. Here we compare two setups, i.e. the reference case with present-day (2014) emissions and a setup with pre-industrial (1750) emissions for anthropogenic and biomass burning sources. Other emissions from natural sources are left unchanged between the two setups to isolate the anthropogenic effect. As mentioned in Sect. 2, radiatively active gases as well as the meteorology are also unchanged from present-day levels, so that the resulting radiative forcings are solely due to changes in the concentrations of aerosols and the resulting cloud modifications. Using prescribed pre-industrial instead of present-day greenhouse gas concentrations may lead to additional changes in the INP effects, due to the climate forcing by anthropogenic greenhouse gases masking the forcing from the INP-cloud interactions. The INP-cirrus effects are shown for present-day and pre-industrial aerosol conditions in Fig. 6. Additionally, the geographic distributions for the pre-industrial case are depicted in Fig. S6 in the Supplement. In general, the INP-effect due to anthropogenic emissions results in lower ice crystal numbers and cloud occurrences, and a larger reduction of homogeneously formed ice crystals, compared to the pre-industrial times. This reduces the warming effect due to cirrus clouds and consequently results in a larger, i.e. more negative, radiative forcing in the present-day case. The global ERF of anthropogenic INPs is $-29 \pm 33$ mW m$^{-2}$ (confidence level of 92 %), calculated from the difference in the INP-cirrus effect between present-day ($-55 \pm 31$ mW m$^{-2}$) and pre-industrial conditions ($-26 \pm 25$ mW m$^{-2}$; Fig. 6a). Considering the current Intergovernmental Panel for Climate Change (IPCC) best estimate of the total effective radiative forcing due to aerosol-cloud interactions of $-1.0 \pm 0.7$ W m$^{-2}$ (Arias et al., 2023), the anthropogenic INP-cirrus effect simulated here ($-29 \pm 33$ mW m$^{-2}$) is very small. In the Northern Hemisphere, the present-day ERF is about twice as large as the pre-industrial value (Fig. 6a), as a result of the strong anthropogenic emissions in this region. Additionally, the geographic distributions of cloud frequency changes between present-day (Fig S3i) and pre-industrial (Fig S5i) show, that the regions with reduced cloud frequency are more pronounced in the present-day case (especially in the Northern Hemisphere), while the regions with increased cloud frequency remain mostly unchanged with respect to pre-industrial times. Again, this suggests, that the effect of increased cloud occurrence in the presence of INPs, is already exhausted by natural INPs. The additional anthropogenic INPs further decrease the cloud frequency in specific regions (see Fig. S4i and Fig. S6i in the Supplement). Consequently, the cooling effect due to anthropogenic INPs is more pronounced and shows a larger statistical significance on the Northern Hemisphere, i.e. $-67 \pm 28$ mW m$^{-2}$.

### 4.2 Sensitivity of the INP-cirrus effect to the model representation of the vertical velocity

In the following we discuss the particular role of the updraft speed of air parcels, which control their adiabatic cooling rate, as the temperature decreases during the lifting process. Various studies show that this dynamic forcing is crucial for the competition between heterogeneous and homogeneous freezing (e.g. Kärcher and Lohmann, 2002, 2003; Kärcher et al., 2006; Kärcher, 2022).

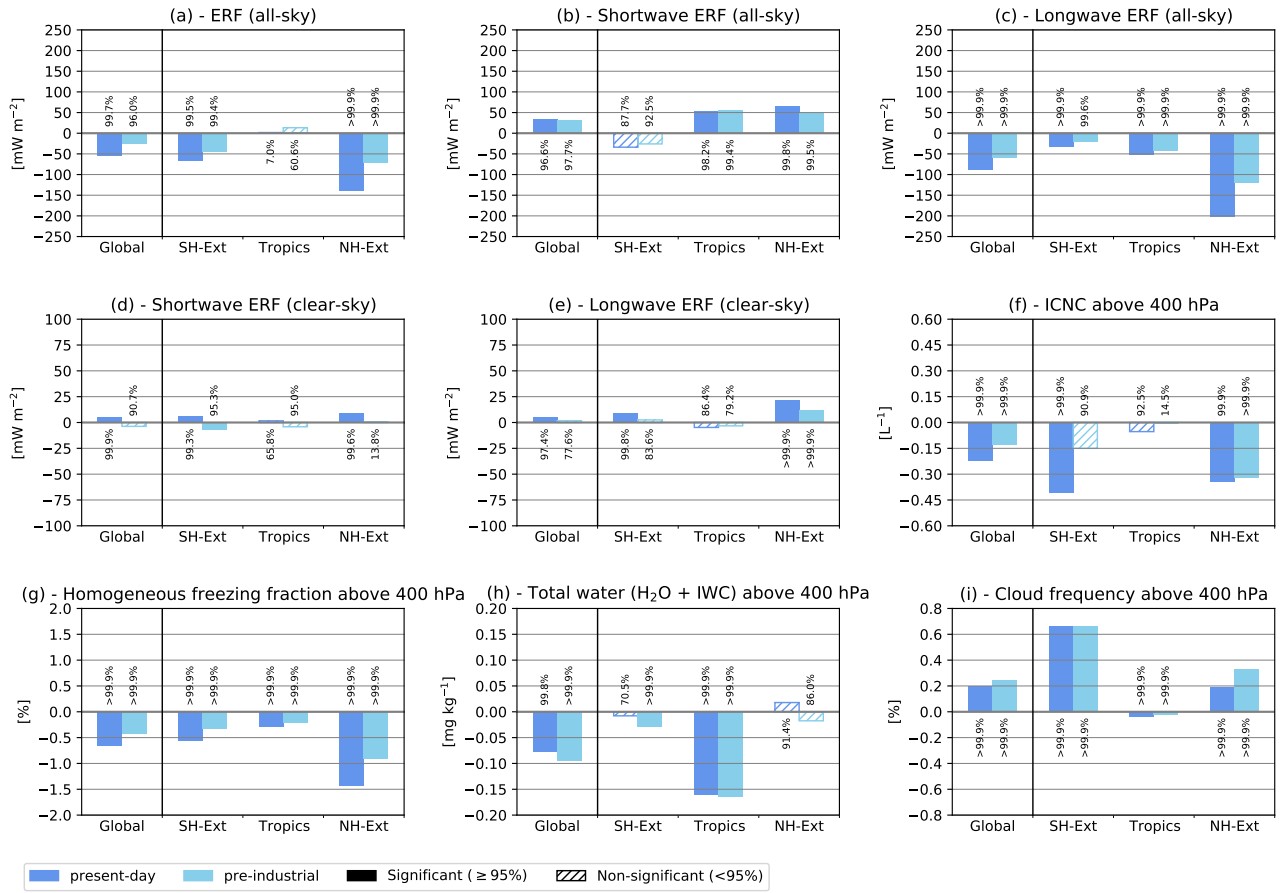

**Figure 6.** As in Fig. 3, but showing the total INP-cirrus effects for present-day (2014) and for pre-industrial (1750) emission conditions. The pre-industrial setup considers emissions for anthropogenic and biomass burning sources representative of the year 1750. All panels show absolute differences with respect to the purely homogeneous freezing case in the two emission setups. Global and latitude-specific, regional differences are shown for (a) total all-sky, (b) all-sky shortwave, (c) all-sky longwave, (d) clear-sky shortwave, (e) clear-sky longwave top-of-the-atmosphere ERFs. Panels (f-i) depict changes in the all-sky ICNC, fraction of homogeneously formed ice crystals, total water (as the sum of water vapour and ice water), and cloud occurrence frequency, all spatially averaged above the 400 hPa level, considering multi-year averages over the simulated period (2001–2010).

To analyse the sensitivity of INP-induced cirrus modifications to the vertical velocities in the model, a simplified representation for the updraft speed is employed here, as also described in Righi et al. (2021), i.e. using a prescribed constant vertical velocity in the cirrus formation process across the whole globe in the range of 1 to $50\,\mathrm{cm\,s^{-1}}$. In comparison, the simulations analysed in the previous sections employed a parametrization for the simulated updraft speeds, including large-scale and sub-grid variability (see Sect. 2 and Fig. S7 in the Supplement showing the global distribution of large scale and sub-grid vertical

velocities). However, treating the vertical velocity in a simplified way (i.e. prescribing a global value) has the advantage that also regions on the globe are taken into account, where no effects would occur in the reference case with parametrized vertical velocities. This is important as the updraft parametrization is subject to uncertainties, e.g. some processes influencing the sub-grid variations of updraft speed may be missing or incorrectly represented (e.g. possible fluctuations due to small-scale, non-orographic gravity waves). Furthermore, the correct representation of vertical velocity fluctuations driven by gravity waves is highly important, as recent studies observed a strong impact of gravity wave temperature perturbations on cirrus cloud occurrence and cirrus properties in the TTL (Kim et al., 2016; Chang and L'Ecuyer, 2020).

Figure 7 shows the sensitivity of the effect of all INPs (compared to pure homogeneous freezing, assuming a central $f_{\mathrm{act}}$ value) to the variation of the prescribed global vertical velocity. Different values of 1, 5, 10, 20, and $50\,\mathrm{cm\,s^{-1}}$ are analysed with respect to changes in the radiative forcing and cirrus properties. For small vertical velocities ($v = 1, 5,$ and $10\,\mathrm{cm\,s^{-1}}$) the global ERF is rather small ($-42$, about 0, and $-31\,\mathrm{mW\,m^{-2}}$, respectively; Fig. 7a). This corresponds to small relative changes in e.g. ice crystal numbers, total water, and cloud occurrences (Fig. 7f, h, i). When increasing the vertical velocity to 20 and $50\,\mathrm{cm\,s^{-1}}$, the ERF increases, i.e. becomes more negative, to $-213$ and $-340\,\mathrm{mW\,m^{-2}}$ (Fig. 7a). Homogeneous freezing is nearly completely inhibited for $v = 1\,\mathrm{cm\,s^{-1}}$ (Fig. 7g), while the homogeneous freezing fraction increases for larger updrafts. Interestingly, the maximal INP-cirrus effect on ICNC and cloud frequency is simulated for $v = 20\,\mathrm{cm\,s^{-1}}$ and decreases again at $50\,\mathrm{cm\,s^{-1}}$, as homogeneous freezing becomes more effective and rapidly consumes the available supersaturated water vapour, as also modelled by Righi et al. (2021). However, this behaviour is not visible in the global ERF, that shows the largest values at $50\,\mathrm{cm\,s^{-1}}$. Comparing the results to the reference case with parametrized vertical velocities, the forcings at smaller prescribed updraft speeds ($\leq 10\,\mathrm{cm\,s^{-1}}$) show similar magnitudes as the reference (simulation F-CEN, Fig. 3a). This indicates that INP-cirrus effects in the reference case are likely controlled by smaller vertical velocities, supporting the findings of Righi et al. (2021) for aviation soot–INP effects. In addition to the global mean values, zonal profiles of the simulated effects are shown in Fig. 8. The results reveal that the simulated radiative forcings are strongest in the extratropics, especially on the Northern Hemisphere. Additionally, the change of sign in the longwave clear-sky ERF from 1 to $5\,\mathrm{cm\,s^{-1}}$ is associated with an increase in total water mainly in the extratroptics (Fig. 8e, h).

In general, the increase of the INP-cirrus-effect with increasing vertical velocity is in line with results from a process-model study by Kärcher et al. (2006). In that study, the authors also showed that for updraft speeds exceeding a certain threshold, homogeneous freezing is gaining increasing importance, leading to a reduction of the effect of heterogeneous INPs on the ICNC, which is in line with the simulated global effects presented here. Regional differences in Fig. 8 (e.g. smaller effects on ICNC and cloud frequency in the tropics) are possibly related to the regional variation in INP number concentrations. The impact of changes in the vertical velocity as presented here is larger compared to a similar study by Righi et al. (2021), where increased forcings of up to a factor of 2 were reported. This is a result of the different investigated effect. While Righi et al. (2021) analyzed the impact of aviation soot INPs (by comparing with a reference case without aviation soot ice nucleation), the present study investigates the effect of all INPs with respect to the case of purely homogeneous freezing.

In an additional set of sensitivity experiments we analysed the impact of scaling the parametrized vertical velocity by factors of 0.2, 0.5, 2, and 5 (See Fig. S8 in the Supplement). Those results show a similar behaviour as for the case of prescribed

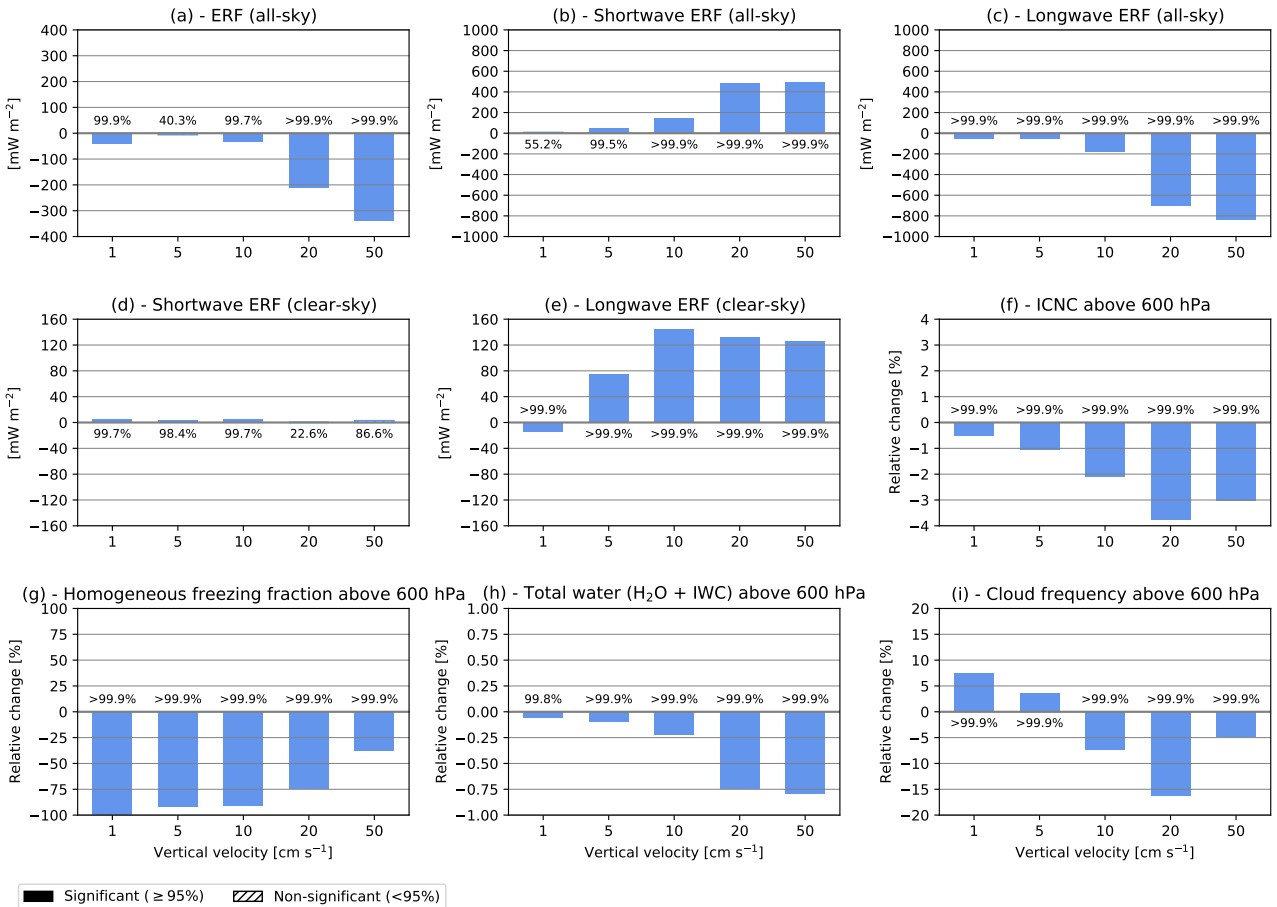

**Figure 7.** Multi-year global averages (years 2001–2010) of changes in (a) total all-sky, (b) all-sky shortwave, (c) all-sky longwave, (d) clear-sky shortwave, (e) clear-sky longwave top-of-the-atmosphere ERFs, and relative changes averaged above the 400 hPa level for (f) all-sky ICNC, (g) the fraction of homogeneously formed ice crystals, (h) total water (as the sum of water vapour and ice water), and (i) cloud occurrence frequency due to the total INP-cirrus effect, considering the difference with respect to the purely homogeneous freezing case, for different values of the vertical velocity, all spatially averaged above the 400 hPa level and over cloudy and cloud-free grid boxes. Confidence levels (in %) are shown for each bar. Significant and non-significant results are represented by filled and hatched bars, respectively.

constant updraft speeds, i.e. more negative ERF and a stronger reduction in ICNC and total water with increasing vertical velocities, albeit showing smaller effects than in Fig. 7 (e.g. a global ERF of $-81\ \mathrm{mW\,m^{-2}}$ for the strongest increase in the vertical velocity). In this study, sub-grid variations of the vertical velocity are represented by the turbulent component of the kinetic energy extended by an orographic gravity wave term (Righi et al., 2020). As explained above, this representation is also subject to uncertainties as certain components of small-scale fluctuations, e.g. non-orographic subgrid-scale gravity waves,

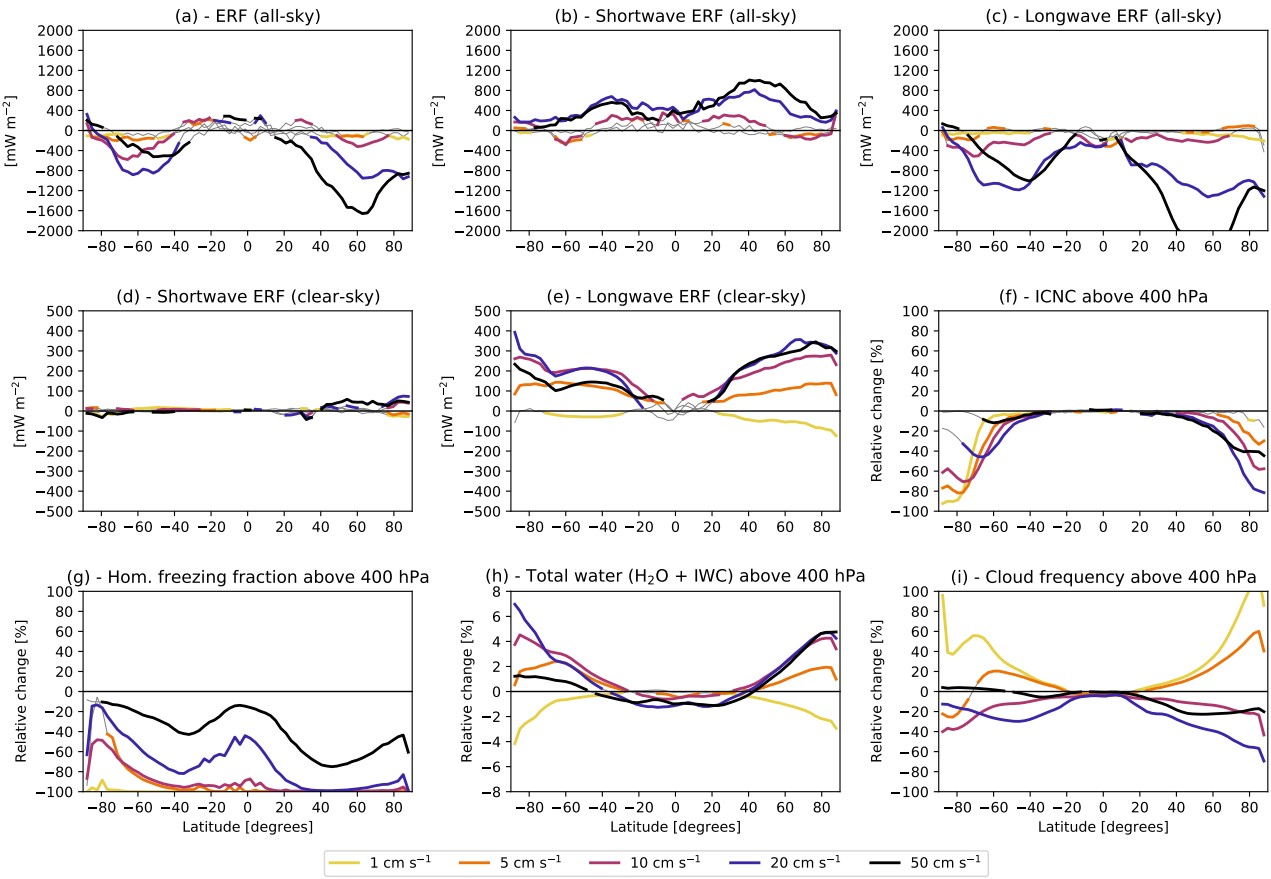

**Figure 8.** As in Fig. 7, but showing zonal averages of changes in (a) total all-sky, (b) all-sky shortwave, (c) all-sky longwave, (d) clear-sky shortwave, (e) clear-sky longwave top-of-the-atmosphere ERFs, and relative changes averaged above the 400 hPa level for (f) all-sky ICNC, (g) the fraction of homogeneously formed ice crystals, (h) total water (as the sum of water vapour and ice water), and (i) cloud occurrence frequency, considering multi-year averages over the simulated period (2001–2010). Non-significant values are shaded in grey.

are not taken into account. In general, parametrizations of sub-grid cooling rate variations suffer from the lack of related observations (e.g. Podglajen et al., 2016). The wide range of simulated radiative effects due to small changes in the updraft speeds shown in this study highlights the importance for future developments in updraft parametrizations in global models.

### 4.3 Impact of model nudging

So far, the results presented above refer to model simulations performed in nudged mode, i.e. relaxing model winds, temperature and surface pressure towards ECMWF reanalyses (see Sect. 2). However, this may affect the INP-induced cirrus cloud and radiation modifications described in the previous sections. Nudging could possibly suppress important feedback mechanisms that would occur in the free-running mode where the meteorology is not influenced by predefined values. However, simulating

in nudged mode results in a more realistic representation of atmospheric dynamics, e.g. by influencing the vertical velocities in the model. In addition, nudged runs suppress differences between simulations due to internal variability (compared to the free-running case), and therefore often require fewer simulated years to achieve statistical significance for the effects investigated here. Here, we compare the INP-cirrus effect between the two cases, i.e. without and with nudging (considering the case with a prescribed vertical velocity of $20\,\mathrm{cm\,s^{-1}}$, i.e. simulations F-CEN-V20 and F-CEN-V20-F, see Table 2). We choose the F-CEN-V20 simulation for this comparison as this run shows large effects on cloud properties and radiation. The resulting global ERF is about 40 % larger (more negative) without nudging ($-290$ compared to $-210\,\mathrm{mW\,m^{-2}}$, respectively), albeit with a lower statistical significance (93.9 % confidence level, see Fig. S9 in the Supplement). The effects on cloud properties, e.g. ICNC or cloud frequency, are generally lower (e.g. about 10 % differences between nudged and free-running setups). Nudging may also impact the simulated aerosol and INP concentrations; however, these changes are mostly small (below 10 %) at cirrus altitudes (data not shown), while larger changes are possible for mineral dust due to changes in the surface wind speeds influencing the wind-driven dust emissions. However, dust emissions have been extensively tuned and evaluated as described in Beer et al. (2020) and rely on the nudged model setup to produce reasonable dust emission values. Therefore, the simulation results performed with the nudged model setup are similar to the free-running case, while some influences of model nudging cannot be excluded. However, as many simulated effects would not be statistically significant in the free-running case, the nudged setup has been applied for most simulations performed in this study.

### 4.4 Uncertainties of simulated INP-cirrus effects

In the previous sections we analysed uncertainties regarding the simulated INP-cirrus effects related to the assumed freezing properties of INPs, the model representation of the vertical velocity, and the impact of model nudging. Further sources of uncertainties are discussed in the following. The diagnostic cloud cover scheme by Sundqvist et al. (1989), adopted in the present model configuration, assumes that a grid box is partly covered by clouds if the grid-mean relative humidity exceeds a critical value, and totally covered if saturation is reached. For the representation of cirrus clouds, supersaturation with respect to ice is allowed; this leads to a cloud cover of 1 when ice nucleation occurs, implying that newly formed cirrus clouds always cover the whole grid box (Kuebbeler et al., 2014). An alternative prognostic treatment of fractional cirrus cloud cover, as proposed by Kärcher and Burkhardt (2008) could reduce this uncertainty and should be the focus of future studies.

A further source of uncertainty in the simulated INP forcings is related to the dependencies on the applied model resolution. As discussed in Sect. 3, the lower horizontal model resolution (T42) compared to previous studies (T63, Beer et al., 2020; 2022) can introduce a positive bias of aerosol concentrations in the upper troposphere. This leads to an increase in INP concentrations in the cirrus regime (of about a factor of 2) and would likely result in larger simulated INP-effects compared to a model setup with a higher horizontal resolution. Overall, the simulated INP concentrations of about 1 to $200\,\mathrm{L^{-1}}$ still agree well with in situ observations and other global model studies as described in Beer et al. (2022). However, direct comparisons of simulated INP number concentrations with in situ observations in cirrus clouds are challenging, as most measurements were performed at lower altitudes and focused on mixed-phase cloud temperatures.

Further uncertainties of the simulated cloud modifications are related to the assumed ice-nucleating properties of the different INPs. These are not clearly resolved by measurements, which usually focus on specific types of particles (e.g. soot particles from a variety of different sources), while information on all possible particles compositions is necessarily limited. Moreover, different particle types of the same aerosol species (e.g. soot) often show a large spread of measured freezing properties (Mahrt et al., 2018). The climate effects of ammonium sulfate and glassy organics are particularly affected by these uncertainties as measurements on their freezing properties are scarce. For example, some studies report a possible reduction in the ice nucleating potential of ammonium sulfate due to coatings with organic material (e.g. Ladino et al., 2014; Bertozzi et al., 2021), which would reduce the impact of ammonium sulfate INPs on cirrus clouds. In addition, the applied phase transition scheme for ammonium sulfate particles considers only ammonium sulfate and neglects different neutralization degrees of other solid sulfate species like ammonium bi-sulfate or letovicite. Recent laboratory experiments by Bertozzi et al. (2024) corroborated the ice-nucleating properties assumed for crystalline ammonium sulfate in the present study but found a reduction in the freezing potential of ammoniated sulfate particles depending on their degree of neutralization. Therefore, future model studies should consider the effect of different neutralization degrees for the representation of atmospheric ammonium sulfate particles and their climate impacts. While recent laboratory studies reported a low ice nucleating potential of glassy organic particles (Piedehierro et al., 2021; Kasparoglu et al., 2022), thus corroborating their non-significant climate impact found in the present study, cloud processing could possibly enhance the ice nucleating abilities of organic aerosols and enhance their impact on cirrus clouds (Kilchhofer et al., 2021). In general, the assumption of a single $f_{\mathrm{act}}$ value for the parametrization of INP-induced heterogeneous freezing in the model is a simplification, as it does not consider increasing activated fractions of INPs during the freezing process.

## 4.5   Effects of INPs with very high ice-nucleating potential

As heterogeneous INPs have the potential to reduce the occurrence frequency and optical thickness of cirrus clouds consequently lowering their warming effect on the global climate, several studies proposed a climate engineering approach to reduce global warming by seeding cirrus clouds with highly efficient INPs (e.g. Mitchell and Finnegan, 2009; Storelvmo et al., 2013; Muri et al., 2014; Gasparini and Lohmann, 2016; Tully et al., 2022). In the present study, the effects of seeding cirrus clouds with different concentrations of very efficient INPs is analysed to estimate the resulting INP effects. Following the procedure described by Gasparini and Lohmann (2016), different INP concentrations (ranging from 0.5 to 100 L$^{-1}$) are prescribed for every model grid box, with a critical freezing supersaturation with respect to ice of $S_c = 1.05$ (the activated fraction is set to $f_{\mathrm{act}} = 1$), and differences compared to the purely homogeneous freezing case are analysed. Vertical velocities are parametrized as in the reference case. By designing the simulation experiments according to the study by Gasparini and Lohmann (2016), we improve the comparability, and aim to explore the robustness of the results presented here by comparing with a similar model study.

Figure 9 shows the global impacts of seeding clouds with different concentrations of highly efficient INPs. In most cases seeding results in a positive global ERF (up to $86\,\mathrm{mW\,m^{-2}}$ for an INP concentration of $100\,\mathrm{L^{-1}}$). Only for the case with a concentration of $10\,\mathrm{L^{-1}}$ a negative forcing could be achieved ($-35\,\mathrm{mW\,m^{-2}}$; see Fig. 9a). The largest INP concentration

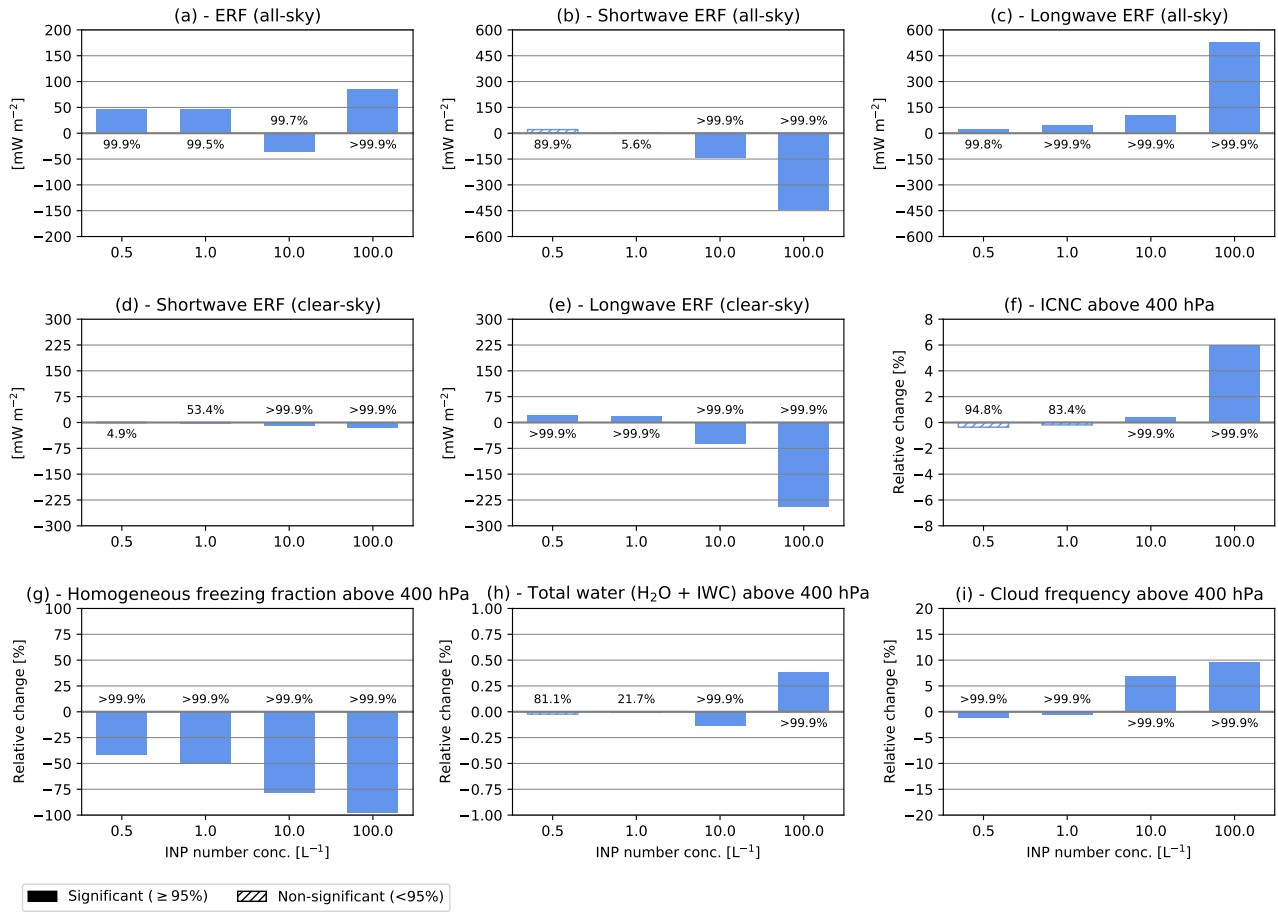

**Figure 9.** As in Fig. 7, but showing multi-year averages (years 2001–2010) of the effect of highly efficient INPs (assuming $S_c = 1.05$, $f_{\mathrm{act}} = 1.0$) for different prescribed concentrations, considering changes in (a) total all-sky, (b) all-sky shortwave, (c) all-sky longwave, (d) clear-sky shortwave, (e) clear-sky longwave top-of-the-atmosphere ERF, and relative changes averaged above the 400 hPa level for (f) all-sky ICNC, (g) the fraction of homogeneously formed ice crystals, (h) total water (as the sum of water vapour and ice water), and (i) cloud occurrence frequency.

also results in a strong increase in ICNC (Fig. 9f), total water (Fig. 9h), and cloud frequency (Fig. 9i) while almost completely inhibiting homogeneous freezing (Fig. 9g). The zonal profiles (Fig. 10) reveal that the positive ERF is related to strong increases in ICNC and cloud frequency in the extratropics. Additionally, the negative longwave clear-sky ERF (Fig. 10e) is driven by reduced total water concentrations in the extratropics.

The heterogeneous freezing effects presented here are likely a result of the very high freezing efficiency assumed for seed
INPs. This assumption implies that heterogeneous freezing is initiated at very low supersaturations with respect to ice and

occurs already at low updraft speeds. For high INP concentrations, ice crystal numbers increase with respect to homogeneous freezing, as a result of the low freezing threshold (i.e. lower $S_c$) compared to homogeneous freezing. This leads to reduced sedimentation due to smaller ice crystals and increased cirrus cloud coverage, which in turn increases the global warming effect due to cirrus clouds. Our results indicate that climate engineering via cirrus cloud seeding risks an overseeding of clouds, subsequently increasing their warming effect, as also argued by Storelvmo et al. (2013) and Penner et al. (2015).

Consistent with the results presented here, Gasparini and Lohmann (2016), employing a similar cirrus cloud scheme, also describe positive radiative effects at large concentrations of seeded INPs, i.e. $490 \pm 240 \, \mathrm{mW \, m^{-2}}$ at 100 INP $\mathrm{L^{-1}}$. Performing the simulations for the seeding concentration of $100 \, \mathrm{L^{-1}}$ in the free-running mode leads to an increased radiative effect and improves the comparison with the results of Gasparini and Lohmann (2016) (see Fig. S8 in the Supplement). For smaller INP concentrations Gasparini and Lohmann (2016) found small net negative forcings, albeit with uncertainty ranges reaching positive values. Recently, Gasparini et al. (2020) showed that cirrus cloud seeding with optimal seeding conditions, i.e. correct INP concentrations, seeding only cirrus clouds during night, could counteract global warming. Also, Tully et al. (2022) showed that choosing larger critical supersaturations $S_c$ for seeding particles could reduce the overseeding effect. However, a recent study by Tully et al. (2023) used a prognostic cirrus seeding aerosol species emitted along aviation soot emissions described strong overseeding and large top-of-atmosphere warming effects even when seeding was restricted to the Northern Hemisphere during winter. Consequently, the potential deleterious effects of overseeding remain, making the feasibility of this climate engineering approach highly uncertain.

## 5 Conclusions

In this study we applied the EMAC-MADE3 global aerosol climate model coupled with a two-moment cloud microphysical scheme to quantify cirrus cloud and radiation modifications due to heterogeneous freezing induced by ice nucleating particles. In addition to the widely investigated INP species, mineral dust and soot, we also analyzed the effects of ammonium sulfate and glassy organic particles, which are only rarely considered as INPs in global modelling studies. Typical mechanisms for INP-induced cirrus effects, as simulated in this study, include the reduction in ice crystal number concentrations, increased sedimentation of larger ice crystals, resulting in a thinning of cirrus clouds, and also regional reductions in cirrus cloud occurrence. On the other hand, cirrus cloud coverage can increase, as INPs freeze at lower supersaturations compared to homogeneous freezing, resulting in earlier cirrus cloud formation during adiabatic cooling in updrafts and consequently increased occurrence frequencies.

The interplay of the above cirrus modifications results in a net negative global radiative forcing due to INP-cirrus interactions, i.e. a cooling effect, of $-28 \pm 21 \, \mathrm{mW \, m^{-2}}$. Assuming an enhanced ice-nucleating potential of INPs (by choosing a larger activated fraction) leads to about a factor of 2 larger INP-cirrus effects and results in an ERF of $-55 \pm 31 \, \mathrm{mW \, m^{-2}}$. The simulated forcings presented here are mostly on the lower end of the range of simulated cooling effects by previous global model studies (e.g. Liu et al., 2012; Wang et al., 2014; Penner et al., 2018; McGraw et al., 2020). The radiative impact of glassy organic INPs simulated here is small and mostly not significant, suggesting that heterogeneous ice-nucleation of

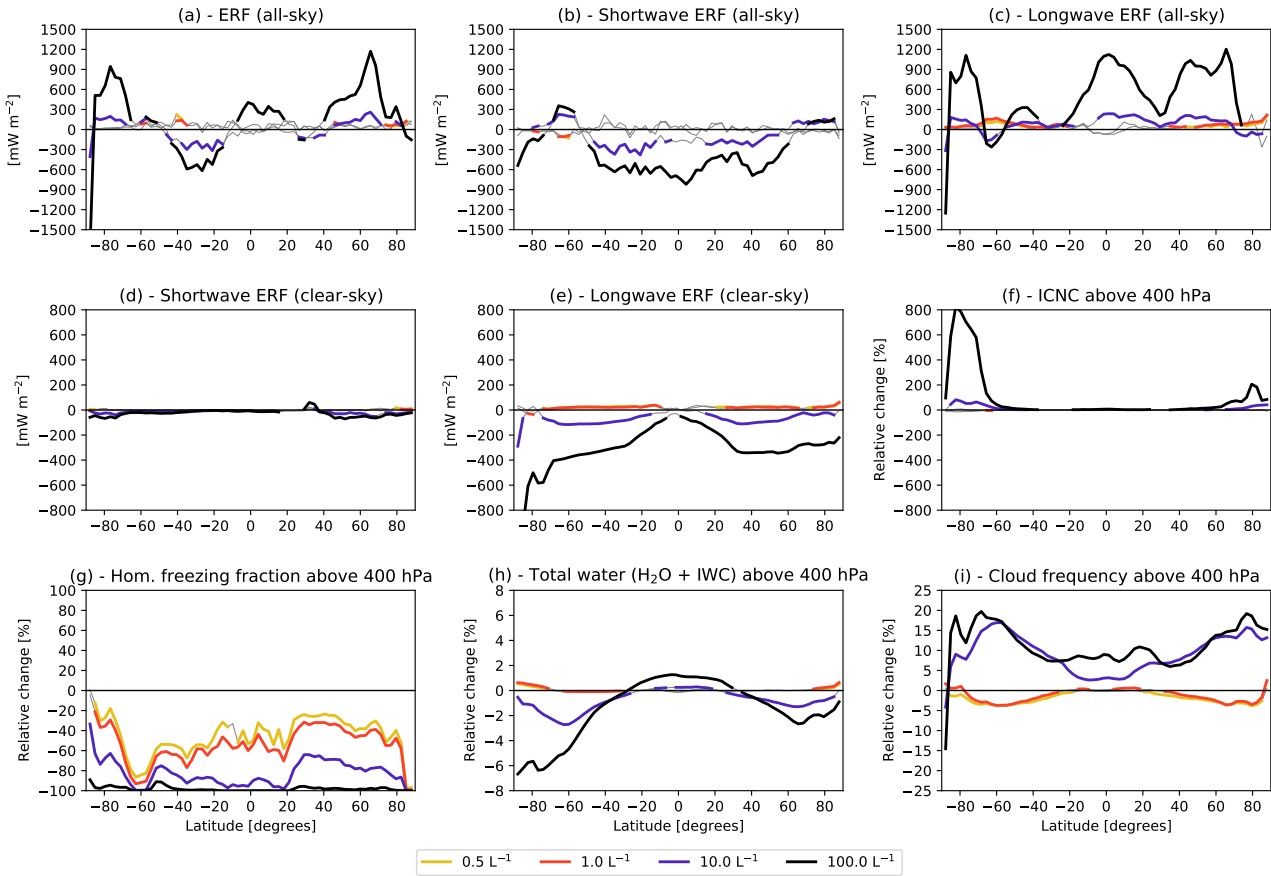

**Figure 10.** As in Fig. 7, but showing zonal averages of changes in (a) total all-sky, (b) all-sky shortwave, (c) all-sky longwave, (d) clear-sky shortwave, (e) clear-sky longwave top-of-the-atmosphere ERFs, and relative changes averaged above the 400 hPa level for (f) all-sky ICNC, (g) the fraction of homogeneously formed ice crystals, (h) total water (as the sum of water vapour and ice water), and (i) cloud occurrence frequency, considering multi-year averages over the simulated period (2001–2010). Non-significant values are shaded in grey.

glassy organic INPs has a negligible role on the global scale. This is in line with low glassy organic INP concentrations
simulated by Beer et al. (2022) and low ice-nucleating abilities reported in recent laboratory studies (e.g. Kasparoglu et al.,
2022). On the other hand, adding crystalline ammonium sulfate to an INP population consisting of mineral dust and soot
results in an additional ERF of $-22 \pm 17$ mW m$^{-2}$, which is nearly as large as the global effect of mineral dust and soot
alone, i.e. $-27 \pm 23$ mW m$^{-2}$. The strong impact of ammonium sulfate is related to its large simulated INP concentrations,
especially on the Northern Hemisphere (Beer et al., 2022). We analysed the effect of anthropogenic INPs, i.e. black carbon and
510 ammonium sulfate related to the combustion of fossil fuels and the use of ammoniacal fertilizers, by comparing the simulated
INP-cirrus effects between present-day (2014) and pre-industrial (1750) conditions. Anthropogenic INP influences are largest
in the Northern Hemisphere and amount globally to an ERF of $-29 \pm 33$ mW m$^{-2}$. However, this anthropogenic INP-cirrus

forcing is small compared with the current IPCC best estimate of the total effective radiative forcing due to aerosol-cloud interactions of $-1.0 \pm 0.7 \, \mathrm{W \, m^{-2}}$ (Arias et al., 2023).

We analyze and discuss the uncertainties regarding the INP-cirrus effects presented here. The use of model nudging can influence the simulated INP-effects due to suppressing feedback mechanisms that would occur in the free-running mode. However, simulation results using the nudging technique are similar to those performed in the free-running mode. Additionally, the use of model nudging is important to achieve statistically significant results. We discuss possible model dependencies on the applied model resolution, which can influence the simulated INP concentrations, as well as cloud formation processes. For

example, an increased horizontal grid resolution can lead to reduced INP number concentrations (about a factor of 2 in the cirrus regime). Therefore, the impact of the applied model resolution on the resulting climate forcing due to INPs should be the focus of future studies.

In additional sensitivity experiments we analyse the role of highly efficient INPs, e.g. proposed for cirrus cloud seeding as a means to reduce global warming by climate engineering. Our results show that this approach often results in the contrary effect,

i.e. positive effective radiative forcings of up to $86 \, \mathrm{mW \, m^{-2}}$, depending on the number concentration of INPs. Choosing too large INP concentrations often risks an overseeding of the clouds and results in strongly increased cirrus occurrence, as also stated by Storelvmo et al. (2013), Gasparini and Lohmann (2016), and Gasparini et al. (2020), making the feasibility of this climate engineering approach highly uncertain.

The INP-cirrus effects shown here are strongly dependent on the representation of the vertical velocity in the model, which

controls the adiabatic cooling rate during the uplifting of air parcels. By performing sensitivity experiments with a prescribed, geographically uniform, vertical velocity, we show that increasing the vertical velocity results in larger simulated INP-cirrus effects, e.g. global ERFs of $-42 \, \mathrm{mW \, m^{-2}}$ (at $1 \, \mathrm{cm \, s^{-1}}$) to $-340 \, \mathrm{mW \, m^{-2}}$ (at $50 \, \mathrm{cm \, s^{-1}}$). This strong sensitivity to the prescribed vertical velocity highlights the crucial role of the dynamic forcing for the simulated climate impact of INPs. The larger impact of changes in the vertical velocity with respect to the study by Righi et al. (2021), where increased forcings of up

to a factor of 2 were reported, is due to the differences in the investigated effects. While Righi et al. (2021) analyzed the impact of aviation soot INPs under varying updrafts, this study investigates the effect of all INPs with respect to the case of purely homogeneous freezing. The present study corroborates the sign of the INP-cirrus effect, i.e. a cooling impact, as simulated in most previous global model studies (Penner et al., 2018; Zhu and Penner, 2020; McGraw et al., 2020; Righi et al., 2021). However, our results disagree with some previous studies in terms of the magnitude of the simulated climate forcings. This

highlights the still large uncertainties in simulated INP-induced cirrus effects and the need for detailed analyses to investigate the causes of these large model diversities. Importantly, the present study suggests a strong contribution of ammonium sulfate to the simulated INP-cirrus effect, also regarding its influence on the anthropogenic INP-cirrus impact, and should be addressed in future studies.

In the future, a more precise knowledge of the ice-nucleating properties of the different INPs from measurements, especially

regarding ammonium sulfate and glassy organics, could help to further constrain their global impact in climate simulations. Additionally, research on the role of cloud-processing of INPs and its potential to enhance their ice nucleating abilities needs to be continued. Considering the strong impact of ammonium sulfate INPs shown here, dedicated observations on atmospheric

ammonium sulfate particles and further model studies could help to better constrain their climate impacts. With regard to the strong sensitivity of the simulated INP-cirrus effects on the vertical velocities in the model, dedicated field observations on atmospheric updraft speeds could help to improve the representation of vertical velocities in the model. A detailed combination of model–observation analyses on INP-induced cirrus modifications employing different measurement techniques like in-situ aircraft observations, lidar measurements, and satellite remote sensing could help to constrain critical model parameters and in turn improve the simulated INP-cirrus effects. Additionally, recent advancements in the representation of INP-induced cirrus formation (e.g. Kärcher, 2022), e.g. by explicitly following the freezing process along the whole activation spectrum instead of using sharp freezing thresholds, could help to improve the resulting simulated climate impacts. Regarding the wide range of simulated INP-cirrus effects from global model studies, performing commonly designed experiments in the context of intercomparison projects could increase the models inter-comparability and could help to better understand the diversity of simulated results. Notable examples of such inter-comparison exercises are the CMIP activities (e.g. Eyring et al., 2016) or the AeroCom community (e.g. Gliß et al., 2021; see also https://aerocom.met.no/, last access January 22, 2024). Recently, Righi et al. (2021) and Kärcher et al. (2023) presented model analyses on the aviation soot–cirrus effect. The addition of ammonium sulfate and glassy organics as background INPs could lead to additional competitions between the different INP species and impact the simulated aviation soot effect, which should be re-evaluated in the future.

*Code and data availability.* MESSy is continuously developed and applied by a consortium of institutions. The usage of MESSy, including MADE3, and access to the source code is licensed to all affiliates of institutions which are members of the MESSy Consortium. Institutions can become members of the MESSy Consortium by signing the MESSy Memorandum of Understanding. More information can be found on the MESSy Consortium Website (http://www.messy-interface.org, last access: January 22, 2024). The model configuration discussed in this paper has been developed based on version 2.54 and is part of the current EMAC release (version 2.55). The exact code version used to produce the result of this paper is archived at the German Climate Computing Center (DKRZ) and can be made available to members of the MESSy community upon request. The model setup and the simulation data analysed in this work are available at https://doi.org/10.5281/zenodo.10276710.

*Author contributions.* CB conceived the study, implemented the model developments concerning the representation of vertical velocities and seeded INPs, designed and performed the model simulations, analysed the data, evaluated and interpreted the results, and wrote the paper. JH contributed to conceiving the study and to the model developments, the model evaluation, the interpretation of the results, and to the text. MR assisted in preparing the simulation setup, helped designing the evaluation methods, and contributed to the model developments, the interpretation of the results and to the text.

*Competing interests.* The authors declare that they have no conflict of interest.

*Acknowledgements.* The model simulations and data analysis for this work used the resources of the Deutsches Klimarechenzentrum (DKRZ) granted by its Scientific Steering Committee (WLA) under project ID bd0080. We are grateful to Elena De La Torre Castro (DLR, Germany) for her comments and suggestions on an earlier version of the manuscript; to George Craig (LMU, Germany), Patrick Jöckel, Robert Sausen, and Helmut Ziereis (DLR, Germany) for helpful discussions. We are grateful for the support of the whole MESSy team of developers and maintainers.

This study has been supported by the DLR transport programme (projects "DATAMOST", "Global model studies on the effects of transport-induced aerosols on ice clouds and climate", "VEU2", and "TraK"), the DLR space research programme (projects "KliSAW" and "MABAK"), the German Federal Ministry for Economic Affairs and Climate Action – BMWK (project "DoEfS"; contract no. 20X1701B), and the Initiative and Networking Fund of the Helmholtz Association (project "ESM").

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
