# Peer review of "Impacts of ice-nucleating particles on cirrus clouds and radiation derived from global model simulations with MADE3 in EMAC"

_EGUsphere, 2023_

## Author Response (AR1)

**Impacts of ice-nucleating particles on cirrus clouds and radiation derived from global model simulations with MADE3 in EMAC**
*C. Beer, J. Hendricks and M. Righi*

**Replies to referee comments**

We are grateful to the reviewers for their insightful comments and constructive criticism, which greatly helped us to improve the manuscript. Please find our point-by-point reply below (reviewers' comments are marked in blue, authors' reply in black, and text quotes in *"italic red"*). The line numbers mentioned in the replies refer to the first submission of the manuscript.

Thanks to the valuable comments by the reviewers we included an additional section at the end of the methods section to address the dependencies on the applied model resolution in more detail. As described in the following, we also adapted the text in several places and included some new figures to the Supplement to support the discussion.

**Reply to Reviewer #1**

**Major comments**

1. Line 9: Please try to better motivate the assumption underlying the sensitivity experiments that yield the -55 mW m⁻² radiative effect and to explain the physical reasoning behind this assumption in the main text, perhaps around Lines 115ff.

Thank you for the suggestion. We added the following to the text:

*Line 9: "…, to explore the range of possible forcings due to uncertainties in the freezing properties of INPs, …"*

*Line 117: "With this sensitivity experiment we aim to explore the range of possible forcings due to uncertainties in the freezing properties of INPs…"*

2. Line 16: "one order of magnitude": Please discuss this in the light of Righi et al. (2021) in the results or the conclusions section (e.g. around Line 436). A quick and superficial look only at the abstract of Righi et al. (2021) suggests that the difference in the present study may be much larger here. Righi et al. (2021) mention a factor two.

Righi et al. (2021) analyzed the effect of aviation soot INPs (i.e. comparing a simulation with aviation soot ice nucleation against a simulation without aviation soot INPs). This is different from the total INP effect shown here (comparing a simulation with all INPs against a simulation with only homogeneous freezing). Hence, the simulated effects are different and also the range of forcings regarding the sensitivity to the vertical velocity.

We added the following to the text (line 434): *"The larger impact of changes in the vertical velocity with respect to the study by Righi et al. (2021), where increased forcings of up to a factor of 2 were reported, is due to the differences in the investigated effects. While Righi et al. (2021) analyzed the impact of aviation soot INPs under varying updrafts, this study investigates the effect of all INPs with respect to the case of purely homogeneous freezing"*

3. Line 40: Please specify what you mean by "very large". In line 218, you state that it is on the order of -100 mW m⁻². Providing a typical range (possibly mentioning the magnitude of the outliers) may be better. Regarding the magnitude of the anthropogenic forcing due to INP, I agree with your assessment in Line 262, which is repeated in Lines 421f. Although you are not comparing the same quantities in Line 40 and 262, I think there may be an inconsistency hidden in the qualitative assessments in Lines 40 and Line 262. Please consider rephrasing your statement in Line 40 and/or explain relative to what effect an effect on the order of -100 mW m⁻² is very large. Please also check that your recommendation in Line 17 is consistent with your overall assessment. I think that repeating the discussion in Line 262 in lines 420f was a good idea. I think that this comparison is important for understanding the results of this study in the context of the existing literature.

Thank you for pointing this out. We adapted the text accordingly (line 40): *"Estimates range from statistically insignificant effects (Hendricks et al., 2011; Gettelman et al., 2012), to negative forcings ranging from a few −10*

*to several −100 mW m⁻² (Liu et al., 2012; Zhou and Penner, 2014; Penner et al., 2018; Zhu and Penner, 2020; McGraw et al., 2020; Righi et al., 2021), or even positive forcings of the order of 100 mW m⁻² (Liu et al., 2009)."*

4. In Beer et al. (2020), the authors evaluated simulated dust number concentrations for the upper troposphere at different resolutions with observations. They found very large differences in upper tropospheric dust concentrations depending on resolution (possibly partially due to different tunings), with very large overestimates of dust number concentration relative to observations in some low resolution setups. Such large differences could be expected to influence the sensitivity to anthropogenic INPs found in this study. At the moment, the authors point to Beer et al. (2020) for a model evaluation. However, as far as I can see, Beer et al. (2020) did not evaluate the setup used in the present manuscript. Because of the strong resolution dependence, an evaluation for the resolution (and the tunings) used in this study would, however, be important. The authors should either point the readers to a published evaluation or else provide an evaluation of upper tropospheric dust number concentrations. The authors should also discuss uncertainties related to dust number concentration and/or point the reader to a published discussion. I think that this becomes especially important when looking at radiative effects of anthropogenic INPs.

Thank you for pointing this out. We added a new section (prior to the results section) on model resolution dependencies, where we discuss this issue and present the global distribution of INPs as shown in Beer et al. (2022) but for the different model resolution applied here. We also added a plot similar to Fig. 5 of Beer et al. (2020) to the Supplement, comparing aerosol number concentrations with observations for the different model resolutions. In general, the INP concentrations are about a factor of 2 larger for the T42L41 model resolution, which is applied here, compared to the T63L31 resolution. This could imply that the climate impacts of the INPs estimated in this study are larger than in the higher resolution case. We also mention these resolution dependencies in the section discussing uncertainties:

*"A further source of uncertainty in the simulated INP forcings is related to the dependencies on the applied model resolution. As discussed in Sect. 3, the lower horizontal model resolution (T42) compared to previous studies (T63, Beer et al., 2020; 2022) can introduce a positive bias of aerosol concentrations in the upper troposphere. This leads to an increase in INP concentrations in the cirrus regime (of about a factor of 2) and would likely result in larger simulated INP-effects compared to a model setup with a higher horizontal resolution."*

5. More generally, the authors should briefly explain why they chose such a low horizontal resolution (perhaps because it allowed them to perform many sensitivity runs) and also discuss how this choice could affect their results.

Thank you for mentioning this. The lower horizontal model resolution applied here was necessary due to the computational resources of the many simulations performed in this study. Additionally, we rely on the extensive model tuning of cloud and radiation properties (as described in Righi et al., 2020; 2021) that has been performed for the model resolution applied here. We mention and discuss this in the additional section on model resolution impacts (see comment above).

6. Line 330: Nudging may also suppress responses to changes in INP concentration. Please discuss.

Indeed, nudging may also suppress responses to changes in aerosol and INP concentrations. See the figure below, comparing black carbon and mineral dust INP number concentrations between the free-running and the nudged case. The larger changes for mineral dust (especially in the lower model levels) are a result of the larger surface wind speeds influencing the wind-driven dust emissions. We mentioned this in the text (line 332): *"Nudging may also impact the simulated aerosol and INP concentrations; however, these changes are mostly small (below 10 %) at cirrus altitudes (data not shown), while larger changes are possible for mineral dust due to changes in the surface wind speeds influencing the wind-driven dust emissions. However, dust emissions have been extensively tuned and evaluated as described in Beer et al. (2020) and rely on the nudged model setup to produce reasonable dust emission values."*

[Figure]

Figure 1: Relative zonal mean differences (in %) between a free-running and a nudged simulation (simulated period 2001-2010), regarding black carbon (left) and mineral dust (right) INP number concentrations.

7. Line 333: I would argue that 40% larger does not automatically imply "comparable". Please discuss. If at all feasible, I would strongly recommend to run several additional free running simulations with small perturbations in order to check whether some of the difference could be due to internal variability. Something like slightly changing the stratospheric diffusion (e.g. enstdif=0.99, enstdif=1.01, etc. in ECHAM) during the first month of spin-up or so would be enough. This will give you a range of values for the free running runs. If the value for the nudged run is somewhere near the range from the free-running ensemble (but not necessarily near the average), this would then be sufficient to imply "comparable".

Thank you for pointing this out. We changed the text accordingly (line 333): *"Therefore, the simulation results performed with the nudged model setup are similar to the free-running case, while some influences of model nudging cannot be excluded. However, as many simulated effects would not be statistically significant in the free-running case, the nudged setup has been applied for most simulations performed in this study."*

Additional model simulations are, however, not feasible as they would not be consistent with the simulations presented here, which have been performed on the now decommissioned DKRZ-supercomputer "Mistral". We will however consider this suggestion in future studies with this model configuration.

8. Sect. 3.4 discusses uncertainties but does not present results. I think this discussion should be merged with the introduction and the conclusion section. Especially the uncertainties related to ammonium sulfate should be discussed in the in a prominent place so that this discussion becomes difficult to overlook. The information in Line 342 regarding cloud cover parametrization may also fit into the methods section. I suggest to present this particular piece of information prior to the results.

Thank you for this suggestion. We think that discussing possible sources for uncertainties in a dedicated section makes these issues more visible. As the discussed uncertainties are directly related to the results presented before, we think this section is best located after the results section. However, we followed your suggestion and mention some of these aspects already in the introduction and the methods sections:

*(line 71) "… We discuss possible sources of uncertainties in the simulated INP-effects, e.g. due to the use of model nudging, dependencies on the applied model resolution, and model assumptions regarding the parametrization of INP-cirrus interactions."*

*(line 117) "… We discuss possible sources of uncertainties influencing the simulated results due to the applied cirrus cloud parametrization in Sect. 4.4."*

9. Line 369: Why is the purely homogeneous case used as a reference here?

The analysis regarding the effects of highly efficient INPs (proposed for cirrus seeding) was designed according to a previous model study by Gasparini and Lohmann (2016), that applies a similar model setup and cirrus cloud scheme. In order to facilitate the comparison, we choose the homogeneous freezing case as a reference, as this results in larger effects with increased statistical significance. The aim was to explore the robustness of the results presented here (also regarding the impacts of INPs in general) by comparing with a similar model study.

10. Sect. 3.5. What is the added value compared to previous studies? Please explain.

Thank you for the comment. This section analyzing the effects of highly efficient INPs (proposed for cirrus seeding) was also intended to explore the robustness of the presented results (also regarding the impacts of INPs in general) by comparing with a similar model study (see comment above).

We mention this aspect in the beginning of Sect. 3.5 (line 369): *"… By designing the simulation experiments according to the study by Gasparini and Lohmann (2016), we improve the comparability, and aim to explore the robustness of the results presented here by comparing with a similar model study."*

11. Lines 397-406: Instead of providing a belated introduction, the conclusion section should highlight novel findings. Read together, the introduction and the conclusion section should clearly explain how these new results enhance our knowledge from previous studies. The next paragraph (Lines 407 to 423) nicely summarizes and explains the results in the context of previous studies.

This paragraph is designed to give the reader a short overview of the topic, the method, and the typical mechanisms for INP-induced cirrus effects, as simulated in this study. However, we shortened it by removing the sentence (line 400) *"In general, … cirrus clouds."*

12. Line 491: Does this mean your next manuscript will be based on changing the assumption of a purely homogeneous case in Line 369? I think that adding this analysis to the present manuscript could potentially provide the added value compared to previous studies that I currently have trouble finding in Sect. 3.5.

We do not intend to explore the topic of cirrus cloud seeding further, also regarding the previous publications on that topic, which state that cirrus cloud seeding is not a feasible approach for geoengineering (e.g. Tully et al., 2023). Also, this analysis regarding seeding INPs was mainly intended to analyze the robustness of the simulated results presented here by comparing with a study employing a similar model system (see also our reply and text changes mentioned above for the 10[th] comment).

**Other comments:**

1. Line 12: in Line 9 you report two values for assuming a smaller and a larger ice nucleating potential. In line 12 you report only one value. I think you should clarify in line 12 whether the value in line 12 was derived assuming a larger or a smaller ice nucleating potential.

Thank you for pointing this out. We changed the text accordingly: *"… -29 mW m$^{-2}$, assuming a larger ice-nucleating potential of INPs"*

2. Line 124: "... due to the uncertain freezing properties of aviation BC" -> I don't understand the logic behind this argument. Please explain.

As the aviation soot effect is not the main focus of this paper, and the freezing properties of soot particles in general, but especially aviation soot, are very uncertain, we assume the same freezing properties for aviation soot and soot from other sources (since the model is able to distinguish between them).

3. Line 146: Does "possible" imply realistic? Please explain or rephrase.

Thank you for pointing this out. "Possible" implies updraft speeds that might occur in the atmosphere. We changed the text to: *"… to explore the full range of updraft speeds typically occurring in the atmosphere (Podglajen et al., 2016; Barahona et al., 2017)."*

4. Line 199: "[P]ossibly due to increased cloud lifetime effects in the presence of INPs": I think you explain what you mean in Lines 236f and also in Lines 405f. Please explain already here. If you are aware of existing references for this, please mention, perhaps already in the introduction. Unless you are very sure about your suggestion and

can explain it better, you could perhaps consider rephrasing this and simply write more frequent (or increased) cloud formation due to INPs instead of increased cloud lifetime effects, or else write more frequent (or increased) cloud formation due to INPs and possibly increased cloud lifetime. The best may be to cite literature. I think you may find something for both statements, although I am not sure, and different studies may yield qualitatively different results.

Thank you. We rephrased the text and included some references:

*(line 199) "… possibly due to more frequent cloud formation or increased cloud lifetimes in the presence of INPs, as a result of the INPs initiating ice formation earlier, i.e. at lower critical supersaturations, compared to homogeneous freezing. Both pathways, i.e. a decrease or an increase in cirrus cloud occurrence due to INPs, have been reported in previous modelling studies depending on the ambient atmospheric conditions and the availability and ice-nucleating properties of INPs (Kuebbeler et al., 2014; Gasparini and Lohmann, 2016; McGraw et al., 2020). Notably, this contributes to the challenge in quantifying the radiative impacts, due to the high variability of the different effects."*

5. Line 254: Please re-iterate that the concentrations were unchanged from near present-day (2014) levels and perhaps also state this even more clearly already in Sect 2. Around line 254, you could briefly mention masking effects by anthropogenic greenhouse gases. I think that if you used 1850 instead of near present-day greenhouse gas concentrations, the INP effect may increase slightly because of anthropogenic greenhouse gases masking the cloud changes. I also suggest to change "present-day and pre-industrial conditions" to "present-day and pre-industrial aerosol conditions" in line 254.

Thank you for mentioning this. We agree and changed the text accordingly:

*(line 252) "… radiatively active gases as well as the meteorology are also unchanged from present-day levels, so that the resulting radiative forcings are solely due to changes in the concentrations of aerosols and the resulting cloud modifications. Using prescribed pre-industrial instead of present-day greenhouse gas concentrations may lead to additional changes in the INP effects, due to the climate forcing by anthropogenic greenhouse gases masking the forcing from the INP-cloud interactions. The INP-cirrus effects are shown for present-day and pre-industrial aerosol conditions …"*

6. Line 286: How do these assumptions compare with the default simulation? I suggest to once more point the reader to Righi et al. (2021) and to also add a plot, perhaps of the 95th percentile of total vertical velocity (resolved plus parameterized) to the Supplement.

Thank you. We mentioned this in the text and included an additional figure in the Supplement.

*(line 277) "… , as also described in Righi et al. (2021), …"*

*(line 279) "… (see Sect. 2 and Fig. S7 in the Supplement showing the global distribution of large scale and sub-grid vertical velocities)"*

7. Line 436: Could you please discuss potential reasons for the disagreement with the statement in the abstract of Righi et al. (2021)? In the abstract Righi et al. mention a factor two.

See our reply to the 2. major comment above. We explained the differences with respect to Righi et al. (2021) and adapted the text accordingly.

8. Lines 451f "recent advancements ... (Kärcher, 2022)": please be more specific. I suggest to briefly explain what the improvement that you are referring to consists of.

Thank you. We give an example for an important improvement as described in Kärcher et al. (2022): *"…, e.g. by explicitly following the freezing process along the whole activation spectrum instead of using sharp freezing thresholds, …"*

9. Table 2: Should there be a base run that does not include mineral dust and soot? Or did I misinterpret line 226 and Figure 3?

The base run is the one without INPs (i.e. only homogeneous freezing). Fig. 3 shows the impact of DU and BC as the difference between a simulation including these INPs and a simulation with only homogeneous freezing. We also included the information on the reference cases for all investigated effects in Table 2.

10. Figures 1, 3, etc. show cloud occurrence frequency instead of cloud cover. Can this be justified by the modifications to the cloud cover parametrization mentioned in line 341? Or would it make sense to show cloud cover? Please explain or modify. Is cloud occurrence frequency defined using a threshold value for cloud cover?

We use the term "cloud occurrence frequency" to prevent confusion with the term "cloud cover", which may be interpreted as a two-dimensional, vertically integrated quantity (e.g. maximum random overlap). We calculate the cloud occurrence frequency by taking the average over time and the vertical levels (above 400 hPa) of the four-dimensional (time, lev, lat, lon) cloud cover simulated by EMAC.

**Technical:**

Table 1: Stating that all runs were run for 10 years might be enough. I am not sure you need an extra column in the table for this.

Good point: we removed the column from Table 2 and stated in the caption *"… All simulations cover a 10-year period (2001-2010)."*

Line 4: Does "(aviation) soot" mean "soot including soot from aviation"?

Yes, we changed the text and only mention soot in general: *"… mineral dust, soot, …"*

L. 30: ...influence these climatic impacts significantly... -> please rephrase

We rephrased this: *"INPs contribute to the climate impacts of cirrus by changing…"*

Lines 238, 238: increased cloud frequency -> more frequent cloud formation?

We changed the text to: *"… the effect of more frequent cirrus formation …"*

L. 50: extent of the INP population -> please rephrase

We rephrased this as: *"… our knowledge on the global INP population is still uncertain, … "*

L. 148: provieded -> provided

Corrected

L. 267: This suggests -> Again, this suggests (because of Line 238)

Changed as suggested

L. 315: Please add "as explained above" (because of Line 283) or omit.

Changed as suggested: *"As explained above, …"*

L. 322: Please check that this has indeed been "described above".

We changed this to: *"… described in the previous sections"*

L. 325: create much less internal noise -> suppress differences between simulations due to internal variability

Changed as suggested: *"In addition, nudged runs suppress differences between simulations due to internal variability …"*

L. 495: Please correct "ThePhysical"

Corrected

L. 506: I suggest to cite the final published ACP paper instead of the preprint.

Corrected

**Reply to Reviewer #2**

**Major comments**

The simulations were conducted with a lower (higher) spatial (vertical) resolution than those in the study by Beer et al (2022), which presents a contradiction. Is there any reason to decrease the resolution to 2.8°x2.8°, while a better resolution of 1.9°x1.9° has been addressed and proven to provide better simulated aerosol concentrations in (Beer et al, 2022)? A better representation of the aerosol concentration (BC, dust, and sulfate) is crucial to rely on the estimate of the RF. A sensitivity of your results to spatial resolution have to be addressed.

Thank you for pointing this out. Following a similar remark by Reviewer 1, we added a new section (prior to the results section) on model resolution dependencies, where we discuss this issue and present the global distribution of INPs as shown in Beer et al. (2022) but for the different model resolution applied here. We also added a plot similar to Fig. 5 of Beer et al. (2020) to the Supplement, comparing aerosol number concentrations with observations for the different model resolutions. In general, the INP concentrations are about a factor of 2 larger for the T42L41 model resolution, which is applied here, compared to the T63L31 resolution. This could imply that the climate impacts of the INPs estimated in this study are larger than in the higher resolution case.

The choice of a lower horizontal resolution was necessary due to constraints in computing resources to realize the large number of sensitivity experiments performed in this study. Also, we rely on the extensive model tuning of cloud and radiation properties performed by Righi et al. (2020, 2021), applying the T42L41 resolution. We also mention these resolution dependencies in the section discussing uncertainties:

*"A further source of uncertainty in the simulated INP forcings is related to the dependencies on the applied model resolution. As discussed in Sect. 3, the lower horizontal model resolution (T42) compared to previous studies (T63, Beer et al., 2020; 2022) can introduce a positive bias of aerosol concentrations in the upper troposphere. This leads to an increase in INP concentrations in the cirrus regime (of about a factor of 2) and would likely result in larger simulated INP-effects compared to a model setup with a higher horizontal resolution."*

The spatial resolution can impact also the cloud formation and coverage as well as the representation of vertical updraft, please comment.

Thank you for this comment. We address this issue in the added section about model resolution dependencies. See also the figure below, showing relative differences in simulated cloud frequency and vertical velocities between the two different resolutions (i.e. T42L41 and T63L31):

*"Notably, the model resolution can also influence cloud formation in the model, e.g. via changes in the simulated vertical velocity, which acts as a driver for the supersaturation and hence the ice-nucleation processes. In general, the differences in cloud frequency and vertical velocities between the T42L41 and T63L31 model resolutions are relatively small compared to the differences in INP numbers, i.e. mostly below 50 % in the cirrus regime (data not shown). Nonetheless, the applied model resolution can influence the simulated INP-cirrus effects and this impact should be the focus of future studies."*

[Figure]

Figure 2: Relative zonal mean difference (in %) in simulated cloud frequency and vertical velocity between a simulation with T42L41 and a simulation with T63L31 model resolution (simulated period 2000-2004).

What kind of aerosol feedbacks are activated in the simulation (direct, semi-direct or indirect)? Are the emissions (e.g. biogenic) in your model adjusting to temperatures and wind changes due to feedbacks on meteorology? Please comment.

The model is able to simulate direct, semi-direct and indirect aerosol effects. The emissions of anthropogenic aerosol components are prescribed and do not depend on meteorological parameters. However, the emissions of mineral dust and sea spray aerosols are wind driven and therefore calculated online at every model timestep depending on the ambient wind conditions. Biogenic emissions (e.g. SOA precursors like natural terpenes) are prescribed, similar to anthropogenic emissions. The calculation of the glassy fraction of SOA particles depends on the ambient temperature and humidity.

We also refer to our replies and the respective text changes related to the comments by Reviewer 1 (major comments 6 and 7) regarding the use of model nudging. Nudging could suppress some feedbacks but it is necessary to produce significant results.

You mentioned the mixing state, did you investigate the impact of the different mixing states on the radiative forcing? How you treat the mixing state of BC, mineral dust, organics and sulfate? Addressing the impact of the mixing state on the radiative calculations is a major concern.

The mixing state is indeed a crucial aspect for the ice nucleation processes in the model. The MADE3 aerosol module simulates different aerosol mixing states (soluble, insoluble, mixed). The ice nucleation parametrization calculates the freezing processes (homogeneous nucleation, different heterogeneous freezing modes) depending on these different mixing states. For example, we distinguish between immersion freezing of mixed mineral dust particles and deposition freezing of insoluble mineral dust. The procedure to calculate the INP number concentrations for the different aerosol size modes and mixing states was described in detail in Righi et al. (2020) and Beer et al. (2022). In Beer et al. (2022) we also investigated different mixing states for ammonium sulfate and glassy organic INPs.

We mention the different mixing states in the text and refer to the respective publications (Righi et al., 2020; Beer et al., 2022) for details: *(line 135) "Different mixing states of particles are taken into account for the simulated ice-nucleation processes, e.g. the model distinguishes between immersion freezing of mixed mineral dust particles and deposition freezing of insoluble mineral dust."*

How do you calculate the optical properties? Could you provide a description of how you perform the calculation?

The optical properties of aerosols and clouds are calculated in the EMAC submodels AEROPT and CLOUDOPT (Dietmüller et al., 2016), respectively, according to input from the OPAC database (Optical Properties for Aerosols and Clouds; Hess et al., 1998). The OPAC package uses basic optical properties from Koepke et al. (1997). We mentioned this in the text (line 95): *"The optical properties of …"*

How you define the control cases to compare with your RF results (e.g. Fig. 1 where you compare only with homogeneous freezing)? Adding a section on all the control cases chosen would rather simplify the RF interpretation.

Thank you for this suggestion. We included the information on the reference cases, which consider only homogeneous freezing, as additional entries in Table 2 (highlighted in bold):

The paragraph 3.2 shows the sensitivity to the updraft velocity, compared to (Righi et al, 2021), where a similar model configuration has been chosen (e.g. spatial resolution). I would have expected a more detailed comparison between the two papers, given that the model configuration is similar, covering partially the same study period, with the exception of the inclusion of ammonium sulfate.

Thank you for pointing this out. Righi et al. (2021) analyzed the effect of aviation soot INPs (i.e. comparing two simulations with and without the impact of aviation soot INPs on ice nucleation). This is different from the total INP effect shown here (comparing a simulation with all INPs against a simulation with only homogeneous freezing). Hence, the simulated effects are different and also the range of forcings regarding the sensitivity to the vertical velocity. Following a similar comment by Reviewer 1, we added the following to the text:

*We added the following to the text (line 308): "The impact of changes in the vertical velocity as presented here is larger compared to a similar study by Righi et al. (2021), where increased forcings of up to a factor of 2 were reported. This is a result of the different investigated effect. While Righi et al. (2021) analyzed the impact of aviation soot INPs (by comparing with a reference case without aviation soot ice nucleation), the present study investigates the effect of all INPs with respect to the case of purely homogeneous freezing."*

A section commenting on the source and variability of the number and mass concentration, as well as size distribution of potential INPs (e.g. DU, BC, organics, sulfate) is missing and it would be appropriate to add it in the manuscript in order to compare the magnitude to the radiative forcing. Which is the major contributor between DU, BC, organic and sulfate to the RF? Given the key role of radiative calculations in this manuscript, I believe it would be more suitable to include a thorough discussion of these aspects within the paper.

The underlying distribution of INP number concentrations is indeed a crucial aspect for the resulting INP-cirrus effects. A thorough discussion on global distributions of the different INPs (and the calculations of their number concentrations depending on aerosol sizes and mixing states) has been presented in Beer et al. (2022). Nevertheless, we added a figure showing the global distribution of number concentrations of the different INP species (similar to Fig. 5 of Beer et al., 2022) in the new section discussing the dependency on the model resolution (see comments above).

In Sec. 3.1.1 you compare the impact of sulfate to dust and BC. Did you compare the impact of sulfate compared to aviation soot only?

Thank you for the question. As the aviation soot-cirrus effect is not the focus of the present study, and has been investigated in detail by Righi et al. (2021), we did not compare the impact of ammonium sulfate to the aviation soot only case. Our focus here is on the overall impact of ammonium sulfate (and glassy organics) compared to the usual INP species mineral dust and soot.

Did you validate your simulated concentrations with observations? In the conclusions you highlight the importance of the usage of observations as a constraint to the radiative calculations.

Thank you for pointing this out. In Beer at al. (2022), we did compare the simulated INP concentrations with results from different observational studies, showing in general a good agreement. However, direct comparisons of model results with in situ observations of INP number concentrations in cirrus clouds are challenging, as most measurements were performed at lower altitudes and focused on mixed-phase cloud temperatures.

Additionally, atmospheric observations focusing on specific particle types (e.g. ammonium sulfate) under cirrus conditions would be needed to constrain their impacts simulated here.

We included the following text in the section discussing uncertainties, where we also mention model resolution dependencies: *(line 345) " … Overall, the simulated INP concentrations of about 1 to 200 $L^{-1}$ still agree well with in situ observations and other global model studies as described in Beer et al. (2022). However, direct comparisons of simulated INP number concentrations with in situ observations in cirrus clouds are challenging, as most measurements were performed at lower altitudes and focused on mixed-phase cloud temperatures."*

L.1-4 Please specify the study period.

All simulation experiments presented here cover the 10-year period 2001-2010. However, the actual period is not fundamental for the analyses presented here, as it only concerns the reanalysis data used to nudge the model and also because we focus on long-term, multi-annual means und use prescribed anthropogenic emissions representative of the year 2014. Anyway, we added this information to the text: *"… to quantify the climate impact of INPs on cirrus clouds (simulated period 2001-2010)"*

L.15 Please quantify.

Thank you. We changed the text: *"… results in positive radiative forcings of up to 86 mW $m^{-2}$ depending on the number concentration of seeded INPs."*

L.17 Please detail.

Thank you. We changed the text: *"… resulting forcings increase about one order of magnitude (-42 to -340 mW $m^{-2}$) when increasing the prescribed vertical velocity (from 1 to 50 cm $s^{-1}$)."*

L.35 The definition of "radiative forcing" is missing. I would rather add here a description of what is the definition of radiative forcing. At some point in the manuscript you compare your result with the effective radiative forcing of aerosol-cloud interaction which I assume is a different defined quantity compared to your RF.

Thank you for mentioning this. We included the definition of radiative forcing to the text and also discuss the comparability to effective radiative forcings.

*(line 36) "To characterize the global radiative impact of INPs the term "radiative forcing" (RF) is used, which is defined as the net change of the Earth's energy balance (Ramaswamy et al., 2019), i.e. downward shortwave plus upward longwave radiative flux, due to some imposed perturbation (the impact of INPs on cirrus clouds, in this study)."*

*(line 155) "The radiative forcings reported here explicitly consider the impact of cloud adjustments, as we are employing an aerosol-cloud coupled model. The RF values presented in the following can therefore be regarded as approximations of effective radiative forcings, although the use of model nudging may tend to suppress some feedbacks (see Sect. 3.3)."*

L.35 References to "several global modelling studies" are missing. Please provide references.

These are provided in the following sentences to better differentiate between different model results.

L.39-42 I would rather reformulate evidencing the range of number reported by the references you cited. "very large negative forcing", how much?

Thank you for pointing this out. Following a similar comment by Reviewer 1, we adapted the text accordingly (line 40): *"Estimates range from statistically insignificant effects (Hendricks et al., 2011; Gettelman et al., 2012), to negative forcings ranging from a few −10 to several −100 mW $m^{-2}$ (Liu et al., 2012; Zhou and Penner, 2014; Penner et al., 2018; Zhu and Penner, 2020; McGraw et al., 2020; Righi et al., 2021), or even positive forcings of the order of 100 mW $m^{-2}$ (Liu et al., 2009)."*

L.54 you provide the term "dynamic forcing". Please provide a definition.

The side clause *"… dynamic forcing, induced by the vertical velocities of the air parcels during the freezing process, …"* is describing the term "dynamic forcing".

L.85 how much the resolution is impacting your result? 2.8°x2.8° is very low and may increase the uncertainties on cloud formation, the particle size distribution and updraft velocity. Please comment.

Thank you for mentioning this. The lower horizontal model resolution applied here was necessary due to the computational resources for the many simulations performed in this study. Additionally, we rely on the extensive model tuning of cloud and radiation properties (as described in Righi et al., 2020; 2021) that has been performed for the model resolution applied here. We mention and discuss this in the additional section on model resolution impacts (see also the comments above).

L.86 Is there a particular reason to choose the 2001-2010 study period? It seems that you use the anthropogenic inventory of 2014 to represent the "present day". Please discuss.

The choice of the study period is not very important here, as it only concerns the reanalysis data used to nudge the model and also because we focus on long-term multi-annual means. However, we choose this period as it has been well evaluated in previous studies applying the EMAC-MADE3 model system (Kaiser et al., 2019, Beer et al., 2020, Righi).

L.88 Please specify the resolution of the meteorological data.

Thank you for mentioning this. We included the following information to the text: *"The original reanalysis data with a spectral horizontal resolution of T255 (0.54° x 0.54° ) and a vertical resolution of 60 levels from the ground up to 0.1 hPa have been re-gridded to the model resolution used in this study. The nudging data have a temporal resolution of 6 hours."*

L.94 "and mixing state", which are the specific mixing states you are talking about? External? Internal? Can you provide more information about how you dealing with particle mixing state? This is a crucial part for the radiative calculations.

We included the following information on the representation of different particle mixing states to the text. For a detailed description of MADE3 we are referring to the respective publications (Kaiser et al., 2014; Kaiser et al., 2019):

*"The aerosol components in the aerosol microphysics submodel MADE3 are distributed into nine lognormal modes that represent different particle sizes and mixing states. Each of the MADE3 Aitken-, accumulation-, and coarse-mode size ranges include three modes for different particle mixing states: particles fully composed of water-soluble components, particles mainly composed of insoluble material (i.e. insoluble particles with only very thin coatings of soluble material), and mixed particles (i.e. soluble material with inclusions of insoluble particles)."*

L.96 Please provide a resume of the setup used in this study.

In the beginning of Sect. 2 (lines 81 to 105) we give a short description on the model setup and mention the most important aspects.

L.97 Please specify the anthropogenic and biomass burning inventory resolution. How you regrid the emissions to your low spatial resolution? Please comment.

Thank you for pointing this out. We included the following information to the text: *"Prescribed emission data are provided in a horizontal resolution of 0.5° x 0.5°. The re-gridding to the actual model grid is performed during the model simulation using the algorithm NCREGRID (Jöckel, 2006)."*

L.104 Please detail "all other natural emissions"

Thank you. We added this information: *"…, i.e. biogenic emissions, volcanic emissions, NOx emissions from lightning, emissions of SOA precursors, dimethyl sulfide (DMS) emissions, wind-driven sea-spray emissions."*

L.162-166 are not results. I suggest to include it in the radiative calculation and relative uncertainties section.

The information on how we calculate statistical confidence is strongly related to the results presented in the figures (e.g. Fig. 1). Therefore, we think it should be mentioned here.

L.122 Please details how you are discerning between aviation soot and others sources. Which are the "other sources" you are talking about?

The method for tagging aviation soot emissions has been described in the recent paper by Righi et al. (2021). Soot emissions from the aviation sector are incorporated into a BCtag tracer while all other soot sources, i.e. transport (land-based, shipping, aviation), non-transport (industry, residential heating, etc.), biomass burning are tracked by the standard BC tracer.

L.126 "Natural secondary", biogenic SOA are not only formed by terpene emissions. Can you please detail on which are the species you are taking into account for the SOA formation?

Thank you for pointing this out. Emissions from natural SOA precursors include isoprene, monoterpenes, and other volatile organic compounds (Guenther et al., 1995). We changed the text to *"(e.g. isoprene, monoterpenes, and other volatile organic compounds)"*

L.144 what do you mean by "dynamic influence"?

Influences due to different vertical velocities. This is explained in the text: *(line 137) "… we analyse the influence of variations in the updraft velocities on the INP-cirrus effects."*

L.170-171 Please add a reference.

We refer to *(e.g. Kuebbeler et al., 2014; Penner et al., 2018; McGraw et al., 2020)*

L.173 -28 mWm-2 is the result of a non-significant and a significant RF. Is this result reliable? Please comment.

According to the Student's t-test, that was performed for the total RF, this value of -28 mW m$^{-2}$ is significant (with a 98% confidence level). The two components (shortwave and longwave) don't both need to be significant for the total RF to be significant. Also, the highly significant longwave RF has a larger (negative) value than the shortwave RF and therefore dominates the total RF, i.e. results in a negative forcing.

L.177 the shortwave positive is not significant…

We added *"…, albeit with lower statistical significance"*

L.187-189 Please add references.

We added the following: *"…, as also shown in other global modelling studies (Kuebbeler et al., 2014; McGraw et al., 2020)."*

L.198 "regional reduction in cloud occurrence" not true over the Equator, please comment.

This statement is referring to Fig. S1 in the Supplement showing that regions of strong longwave cooling coincide with regions with reduced cloud frequency. The tropics usually show only small and non-significant INP-effects on the radiative forcing. This is a general feature for all INP-effects simulated in the present study and may be due to the lower availability of INPs in that region (see Fig. 5 in Beer et al., 2022) or the strong impact of convection resulting in stronger updrafts favoring homogeneous freezing.

We included a comment on the low effect in the tropics: *"The vanishing RF in the tropics may also be due to the strong influence of convection in that region, resulting in enhanced updrafts that are more favourable for homogeneous freezing (Kärcher et al., 2006)."*

L.214 "global cooling" how much? Please quantify.

We wanted to point to the negative sign of the forcing here, i.e. a cooling. We adapted the text: *"In general, the negative sign of the simulated global forcing, i.e. a global cooling, as a result …"*

L.217 "-100mW/m2" to what RF is referred?

The forcing of the order of -100 mW m$^{-2}$ refers to the global RF due to INP-cirrus interactions simulated in previous studies. This is mentioned in the text: *(line 214) "…the global cooling as a result of cloud modifications due to heterogeneous freezing…"*

L.219 Please quantify.

The small simulated forcings in the studies mentioned here are not significant. Therefore, we do not mention a value for the RF here.

L.235 "large number concentration of ammonium sulfate INPs in that region", how the spatial resolution impacts the magnitude of the RF of this result? You are referring to a temporal and spatially different simulation. And what about the non-significant RF at the tropics? Please comment.

Thank you for mentioning this. As described in the added section on model resolution dependencies, the INP concentrations are about a factor of 2 larger in the T42L41 resolution applied here compared with the T63L31 resolution used in Beer et al. (2020). This may also lead to increased radiative forcings due to INPs. However, the statement about "larger number concentrations of ammonium sulfate INPs in that region" (i.e. the Northern Hemisphere)" still holds for the resolution applied here. Regarding the low RF in the tropics, see the comment and the text changes mentioned above.

L.236-237 "Notably…soot", please add a reference

Thank you. We added references to Fig. 4i and Fig. 3i

L.260-262 you compare your results with the results from the IPCC. Are they comparable? RF and ERF are different in the definition.

Thank you for pointing this out. We refer to our reply above commenting on the comparability of the RF values. We included the following to the text: (line 155) *"The RF values presented in the following can therefore be regarded as approximations of effective radiative forcings, although the use of model nudging may tend to suppress some feedbacks (Sect. 3.3)"*

L.269 please provide a reference.

We added references to Fig. S3i and Fig. S5i: *"… decrease the cloud frequency in specific regions (see Fig. S3i and Fig. S5i in the Supplement)."*

L.320-335 is this paragraph necessary? we choose the nudging to improve the simulation.

Nudging could possibly suppress feedback mechanisms that would occur in the free-running mode. This has been analyzed by comparing the INP-effects between these two cases, i.e. nudged and free-running and the results are described in this section. We also refer to our replies to Reviewer 1 (major comment 6. and 7.) and the respective text changes.

L.360 "highly efficient INP", could you please detail what you mean with "highly efficient"?

Thank you. We mean "highly efficient INPs" in terms of INPs with very high ice-nucleating potential (as proposed for seeding cirrus clouds as an approach for climate engineering). We changed the text to *"Effects of INPs with very high ice-nucleating potential"*

L.397-406 This paragraph does not contain conclusions…

We shortened this paragraph by removing the sentence "In general, … cirrus clouds." However, it is designed to give the reader a short overview of the topic, the method, and the typical mechanisms for INP-induced cirrus effects, as simulated in this study.

L.413 horizontal resolution may impact this "low glassy organic INP". Please discuss.

Thank you for pointing this out. We refer again to the newly added section about model resolution dependencies. An increased horizontal model resolution would change the INP concentrations. However, as it would in fact lead to a decrease (about a factor of 2) in INP concentrations, the radiative impact of glassy organic INPs would likely

decrease, so that the statement about a low impact of glassy organic INPs would still be valid at increased model resolutions.

L.437-448 Here you highlight the importance of integrating measurements as a constraint to radiative forcing results. It would have been very interesting to integrate measurements in the current paper. In the manuscript you give an overview of your simulated results, but no validation/constraint with observations is used to validate the order of magnitude of your result. Please discuss.

In Beer at al. (2022), we did compare the simulated INP concentrations with results from different observational studies, showing in general a good agreement. However, direct comparisons of model results with in situ observations of INP number concentrations in cirrus clouds are challenging, as most measurements were performed at lower altitudes and focused on mixed-phase cloud temperatures. Additionally, atmospheric observations focusing on specific particle types (e.g. ammonium sulfate) under cirrus conditions are, to the authors knowledge, not available but would be needed to constrain their impacts in global simulations.

We included the following text in the section discussing uncertainties, where we also mention model resolution dependencies: (line 345) " … Overall, the simulated INP concentrations of about 1 to 200 $L^{-1}$ still agree well with in situ observations and other global model studies as described in Beer et al. (2022). However, direct comparisons of simulated INP number concentrations with in situ observations in cirrus clouds are challenging, as most measurements were performed at lower altitudes and focused on mixed-phase cloud temperatures."

**Minor comments**

L.5 "Several sensitivity experiments"

Changed as suggested.

L.11 Please specify the area of interest for the -29 mWm-2

We changed the text accordingly: *"… -29mW m-2, assuming a larger ice-nucleating potential of INPs."*

L.90 I would rather remove "climatological", 10 years are not such a long period to be considered climatology.

We changed the text as suggested: *"… using prescribed long-term means (2001–2010) of sea-surface temperature …"*

L.93 "nine log-normal modes", please specify the range of diameters.

Typical size ranges of the MADE3 modes are of the order of tens of nanometers (Aitken modes), hundreds of nanometers (accumulation modes), and several micrometers (coarse modes) and are depicted in e.g. Fig. 1 of Beer et al. (2020). MADE3 uses not fixed but dynamical mode sizes, which may change during a simulation and are dependent on the assumption of the emitted and nucleated particle sizes. For details we refer to Kaiser et al. (2014, 2019).

L.38 "sudies" please correct.

Corrected

L.45 "these processes" to "sub-grid processes"

Changed as suggested

L.111 "whole freezing spectrum", please specify what "whole" means.

We changed the text as: "… whole freezing spectrum *(from the freezing onset to the homogeneous freezing threshold) …"*

L.173 the area for the RF of -28 mWm-2 is missing, you mean global?

Yes, we changed the text to: "… global cooling … with an RF of -28 mW $m^{-2}$"

L.204 "In order to explore"

Changed as suggested

**Technical**

*"As in Fig." is recurrent in the manuscript. I would rather suggest to explicitly write a complete description of the figure in the caption.*

As this would lead to much redundant text in the figure captions, and lead to a large increase in the overall length of the paper, we think the present formulation is more appropriate.

*Could you please provide the boundaries of your averaged areas? SH-Ext, Tropics, NH-Ext?*

Thank you. These are provided in the caption of Fig. 1. We also added a reference to the text: *"… (Southern Hemisphere extratropics, tropics, Northern Hemisphere extra-tropics; see the caption of Fig. 1 for details) …"*

*1 "Confidence levels …respectively". This part should go on methods where you detail how you perform the radiative calculations/comparisons*

We think a description on the information about statistical significance, and how it is calculated, is important to interpret the figures and the results. Therefore, this information is presented in the caption of Fig. 1 and in the text (line 162).

*In Tab. 2 the activated fractions are reported as (onset, central). I would rather suggest to quantify the values putting the associated label in parenthesis or vice versa.*

We agree that providing this information is important. However, as it has been presented already in Table 1, and would substantially increase the size of Table 2, if included again there (as all $f_{act}$-values for the different INPs would have to be stated), we think this is not feasible.

*Fig. 3 and Fig. 4 could be merged into one figure.*

Thank you for the suggestion. Although these two figures could be merged in principle, the results consider different reference cases, i.e. the purely homogeneous freezing case for the mineral dust/BC impact, and a baseline with only dust/BC for the case of ammonium sulfate (as also mentioned in the respective sections and figure captions). Therefore, we think these two figures should be shown separately to avoid misinterpretations of the results.

*The magnitude of the clear-sky RF is not readable in the plots.*

The same range of the y-axis for clear-sky RF was chosen for Figures 1, 3, and 4, in order to improve the comparability (also with respect to the all-sky forcings). The respective values are overall very small and mostly not significant.

---

## Author Response (AR2)

**Impacts of ice-nucleating particles on cirrus clouds and radiation derived from global model simulations with MADE3 in EMAC**
*C. Beer, J. Hendricks and M. Righi*
**Replies to referee comments – 2ⁿᵈ phase**

We are grateful and thank the reviewers for their valuable comments in this second round of reviews. Please find our point-by-point reply below (reviewers' comments are marked in blue, authors' reply in black, and text quotes in *"italic red"*).

**Reply to Reviewer #1**

**Minor comments:**

Lines 452-454: I think this sentence could be woven into the introduction (below line 79) in order to better motivate your study. A one sentence summary and conclusion based on this result could be added behind line 519. If I understand it right, your study finds a much smaller effect using more realistic assumptions.

Thank you for the suggestion. We added this aspect to the introduction as suggested:

*Line 79: "By designing the simulation experiments for this analysis according to the study by Gasparini and Lohmann (2016), we improve the comparability and aim to explore the robustness of the resulting quantifications of the INP-cirrus effects presented here by comparing with a similar model study."*

Lines 38-41 and elsewhere: I think it would be better to default to the term ERF instead of RF. Although it is good to mention that nudging may suppress some of the responses to changing INP, I suggest to nevertheless use the term ERF. Because of the absence of water vapor nudging, the nudging here suppresses only responses via dynamics, which I think can have only a very small effect. Therefore, as far as I can see, you are actually computing ERF. (The lack of statistical significance in the free running model is almost certainly due to larger variability between the runs and therefore does not reflect differences in the magnitude of the ERF.)

We agree and use the term ERF instead of RF in the text, while still commenting on possible nudging impacts.

*(line 208) "The radiative forcings reported here explicitly consider the impact of cloud adjustments, as we are employing an aerosol-cloud coupled model. The RF values presented in the following can therefore be regarded as approximations of effective radiative forcings (ERF), although the use of model nudging may tend to suppress some feedbacks (see Sect. 4.3). Therefore, we use the term ERF in the following."*

**Technical:**

Line 184: as presented -> has been presented

We rephrased this sentence to make easier to read:

*"In Fig. S1, we present a comparison of aerosol number concentrations for the different model resolutions, which complements Fig. 5 of Beer et al. (2020) with the results for the T42L41 resolution applied in the present study."*

**Reply to Reviewer #2**

**Specific comments**

- Beer et al, findings refer to a possible larger INP effect due to lower model spatial resolution but they have not quantified the impact of the spatial resolution to the radiative forcing (RF), which is one of the of key aspect of the paper. Would it be possible? How much the RF changes (in %) compared to the higher resolution? This additional information will enable a more comprehensive and accurate assessment of your RF estimations, allowing for a better interpretation of the results.

We agree that the impact of an increased horizontal model resolution on the simulated INP-cirrus effects is an important aspect. However, the model simulations presented here rely on the extensive tuning of cloud and radiation properties performed by Righi et al. (2021). Going to a different model resolution would require a complete retuning of the model, which would in turn require performing a large number of (quite expensive) tuning simulations. This is not possible for the present study. However, we aim to analyze this aspect in the future and already mentioned this in the text:

*(line 199) "Nonetheless, the applied model resolution can influence the simulated INP-cirrus effects and this impact should be the focus of future studies."*

- L. 185 "a factor 2 to 3 larger", which translates to how much RF difference? Could you quantify the term "larger' in terms of RF?

At present we cannot quantify the impact of increasing model resolution on the RF. We aim to analyze this in a subsequent study (see our reply to the comment above).

- I recommend incorporating the spatial resolution sensitivity results into the conclusions section, highlighting the advantages and disadvantages of using this spatial resolution.

Thank you. We follow the suggestion and mention the sensitivity to the model resolution in the conclusion section. We also include some remarks on the use of model nudging in this paragraph.

*(line 508) "We analyze and discuss the uncertainties regarding the INP-cirrus effects presented here. The use of model nudging can influence the simulated INP-effects due to suppressing feedback mechanisms that would occur in the free-running mode. However, simulation results using the nudging technique are similar to those performed in the free-running mode. Additionally, the use of model nudging is important to achieve statistically significant results. We discuss possible model dependencies on the applied model resolution, which can influence the simulated INP concentrations, as well as cloud formation processes. For example, an increased horizontal grid resolution can lead to reduced INP number concentrations (about a factor of 2 in the cirrus regime). Therefore, the impact of the applied model resolution on the resulting climate forcing due to INPs should be the focus of future studies."*

- I would suggest to merge L. 400- 410 ("The resulting global RF…") in section 4.3 on model nudging to Sec 4.4. On the other hand, which is the added value of 391-400 lines?

These lines particularly focus on the impact of model nudging, which could potentially impact the INP-cirrus effects due to suppressing some feedbacks. Section 4.4 discusses further uncertainties that are not analyzed in detail in this study (e.g. by performing sensitivity experiments). Therefore, we think this fits better in Sect. 4.3, as this subsection is specifically dedicated to the impact of model nudging.

- Figure 5 shows the impact of AmSu compared to the "heterogeneous freezing case on DU and BC", did you calculate the AmSu RF with the homogeneous freezing case only?

We chose the case with DU and BC INPs as a reference here, in order to analyze the impact of adding the newly implemented AmSu INPs to the population of DU and BC, which are usually considered as INPs in the cirrus regime by global models. In the same way we analyzed the impact of adding glassy organic INPs to the population of DU and BC. Therefore, we did not quantify the impact of only AmSu compared to purely homogeneous freezing. However, we appreciate this suggestion and may consider it for future analyses.

- L. 297-298 Fig. 5 and Fig. 4 refer to different reference cases and the results are not directly comparable.

We agree that these two results are not directly comparable due to the different baseline cases. For this reason, we chose to show these effects in two different figures. We explicitly mentioned the different baseline in the figure caption and in the text.

*(line 293) "This is calculated as the difference between a simulation including AmSu, DU and BC INPs, and a simulation including only heterogeneous freezing on DU and BC."*

*(Fig. 5 caption) "… calculated from the difference between a simulation including AmSu, DU and BC INPs, and a simulation including only heterogeneous freezing on DU and BC. […] Note the different reference case with respect to Fig. 4."*

- Please correct "As in Fig." recurrent in most of the figures. Consider moving some Figures in the supplementary to lighten the text if necessary, instead of reducing the caption's text. A Figure has to be self-consistent in the caption. You do not discuss all the parameters in the Figures, so please also consider to move some panels to the Supplementary material instead of shortening the caption.

Thank you for the suggestion. We included some additional information in the figure captions.

*(Fig. 3 caption) "… spatially averaged above the 400 hPa level and considering multi-year averages over the simulated period (2001–2010)."*

*(Fig. 4 caption) "… Global and latitude-specific, regional differences are shown for (a) total all-sky, (b) all-sky shortwave, (c) all-sky longwave, (d) clear-sky shortwave, (e) clear-sky longwave top-of-the-atmosphere ERFs. Panels (f-i) depict relative changes in the all-sky ICNC, fraction of homogeneously formed ice crystals, total water (as the sum of water vapour and ice water), and cloud occurrence frequency, all spatially averaged above the 400 hPa level, considering multi-year averages over the simulated period (2001–2010)."*

*(Fig. 5 caption) "… Global and latitude-specific, regional differences are shown for (a) total all-sky, (b) all-sky shortwave, (c) all-sky longwave, (d) clear-sky shortwave, (e) clear-sky longwave top-of-the-atmosphere ERFs. Panels (f-i) depict relative changes in the all-sky ICNC, fraction of homogeneously formed ice crystals, total water (as the sum of water vapour and ice water), and cloud occurrence frequency, all spatially averaged above the 400 hPa level, considering multi-year averages over the simulated period (2001–2010)."*

*(Fig. 6 caption) "Multi-year global averages (years 2001--2010) of changes in (a) total all-sky, (b) all-sky shortwave, (c) all-sky longwave, (d) clear-sky shortwave, (e) clear-sky longwave top-of-the-atmosphere ERFs, and relative changes averaged above the 400 hPa level for (f) all-sky ICNC, (g) the fraction of homogeneously formed ice crystals, (h) total water (as the sum of water vapour and ice water), and (i) cloud occurrence frequency due to the total INP-cirrus effect, considering the difference with respect to the purely homogeneous freezing case, for different values of the vertical velocity, all spatially averaged above the 400 hPa level and over cloudy and cloud-free grid boxes. Confidence levels (in %) are shown for each bar. Significant and non-significant results are represented by filled and hatched bars, respectively."*

*(Fig. 8 caption) "As in Fig. 7, but showing zonal averages of changes in (a) total all-sky, (b) all-sky shortwave, (c) all-sky longwave, (d) clear-sky shortwave, (e) clear-sky longwave top-of-the-atmosphere ERFs, and relative changes averaged above the 400 hPa level for (f) all-sky ICNC, (g) the fraction of homogeneously formed ice crystals, (h) total water (as the sum of water vapour and ice water), and (i) cloud occurrence frequency, considering multi-year averages over the simulated period (2001–2010). Non-significant values are shaded in grey."*

*(Fig. 9 caption) "As in Fig. 7, but showing multi-year averages (years 2001–2010) of the effect of highly efficient INPs (assuming $S_c$ = 1.05, $f_{act}$ = 1.0) for different prescribed concentrations, considering changes in (a) total all-sky, (b) all-sky shortwave, (c) all-sky longwave, (d) clear-sky shortwave, (e) clear-sky longwave top-of-the-atmosphere ERF, and relative changes averaged above the 400 hPa level for (f) all-sky ICNC, (g) the fraction of homogeneously formed ice crystals, (h) total water (as the sum of water vapour and ice water), and (i) cloud occurrence frequency."*

*(Fig. 10 caption) "As in Fig. 9, but showing zonal averages of changes in (a) total all-sky, (b) all-sky shortwave, (c) all-sky longwave, (d) clear-sky shortwave, (e) clear-sky longwave top-of-the-atmosphere ERFs, and relative changes averaged above the 400 hPa level for (f) all-sky ICNC, (g) the fraction of homogeneously formed ice crystals, (h) total water (as the sum of water vapour and ice water), and (i) cloud occurrence frequency, considering multi-year averages over the simulated period (2001–2010). Non-significant values are shaded in grey."*

**Technical comments**

- Table 1 could be removed. You can refer to Table 1 in Beer et al, 2022.

Table 1 shows some additional information that was not shown in the respective Table in Beer et al. (2022). Therefore, we think it is necessary to include this table here.

- I would suggest to rename the Sec. 3 to "Model dependencies and uncertainties"

As this section specifically focuses on the dependency on the model resolution and as additional uncertainties are discussed in Sect. 4.4, we think the naming "Model resolution dependencies" is appropriate here.

- I would rather split Table 2 into two tables: 1) reference table with all the specific reference cases (not only the homogenous freezing), adding the reference label in each figure's caption and 2) the global model simulations performed. This will rather simplify the results interpretation.

Thank you for the suggestion. We agree, that the different reference case for the case of the ammonium sulfate effect should be highlighted. We already mentioned this in the text, but now also state it explicitly in Table 2 (see the text quotes below). Despite the ammonium sulfate effect (and the effect of glassy organics), all INP effects analyzed in the present study consider a baseline with purely homogeneous freezing as a reference. Therefore, we think an additional table is not necessary here.

*(Table 2 caption) "[…] The homogeneous freezing reference cases are marked in bold. An exception are the simulations that consider AmSu (glPOM) in addition to DU and BC (simulations F-CEN-DBA, F-CEN-DBG; see below), which consider the simulation with DU and BC as the reference case (simulation F-CEN-DB). […]"*

*(line 293) "This is calculated as the difference between a simulation including AmSu, DU and BC INPs, and a simulation including only heterogeneous freezing on DU and BC."*

- Figure 5 includes as reference case the "heterogeneous freezing on DU and BC" and not the "only homogeneous freezing", this should be highlighted to avoid misinterpretation and included in the aforementioned reference table.

Thank you for pointing this out. We included an additional note in the figure caption and also mention this in Table 2 (see also our reply to the previous comment).

*(Fig. 5 caption) "… Note the different reference case with respect to Fig. 4."*

- For Figure 7 the reference case for each RF estimation is missing. "Pure homogeneous freezing" should be added to the caption.

Thank you. We added this information to the figure caption.

*(Fig. 7 caption) "Multi-year global averages (years 2001–2010) of changes in […] due to the total INP-cirrus effect, considering the difference with respect to the purely homogeneous freezing case, for different values of the vertical velocity…"*